# Combining Markov and Semi-Markov Modelling for Assessing Availability and Cybersecurity of Cloud and IoT Systems

**Vyacheslav Kharchenko [1], Yuriy Ponochovnyi [2], Oleg Ivanchenko [3], Herman Fesenko [1] and Oleg Illiashenko [1,\*]**

[1] Department of Computer Systems, Networks and Cybersecurity, National Aerospace University "KhAI", 17 Chkalov Str., 61070 Kharkiv, Ukraine
[2] Department of Information Systems and Technologies, Poltava State Agrarian University, 1/3 Skovorody Str., 36003 Poltava, Ukraine
[3] Department of Computer Systems Software, Dnipro University of Technology, Dmytra Yavornytskogo Ave. 19, 49005 Dnipro, Ukraine
\* Correspondence: o.illiashenko@khai.edu

**Abstract:** This paper suggests a strategy (C5) for assessing cloud and IoT system (CIS) dependability, availability, and cybersecurity based on the continuous collection, comparison, choice, and combination of Markov and semi-Markov models (MMs and SMMs). It proposes the systematic building of an adequate and accurate model to evaluate CISs considering (1) continuous evolution of the model(s) together with systems induced by changes in the CIS or physical and cyber environment parameters; (2) the necessity of collecting data on faults, failures, vulnerabilities, cyber-attacks, privacy violations, and patches to obtain actual data for assessment; (3) renewing the model set based on analysis of CIS operation; (4) the possibility of choice and utilizing "off-the-shelf" models with understandable techniques for their development to assure improved accuracy of assessment; (5) renewing the models during application of CIS by time, component or mixed combining, taking into consideration different operation and maintenance events. The results obtained were algorithms for data collection and analysis, choice, and combining appropriate MM and SMMs and their different types, such as multi-fragmental and multiphase models, considering changing failure rates, cyber-attack parameters, periodical maintenance, etc. To provide and verify the approach, several private and public clouds and IoT systems were researched and discussed in the context of C5 and proposed algorithms.

**Keywords:** Markov modelling; semi-Markov modelling; availability; cybersecurity; cloud; IoT

## 1. Introduction

The attractiveness of cloud services and Internet of Things (IoT) systems that integrate with them for industrial and individual customers is due to the possibility of deploying scalable capacities and the availability of isolated virtual resources. The demand for cloud services is growing every year, and the most important characteristics of these services are given in the documents of leading institutions, namely NIST, ENISA, IEEE, etc. [1–3]. The mentioned scalability, as well as support for virtualization functions, efficient use, flexibility and, under the first conditions, failure and disaster resistance, provide significant advantages in comparison with grid computing systems, supercomputers, etc.

The growing need for the use of web technologies is due to the emergence of new technological developments for greater flexibility and availability at a reduced cost. An important factor for the effective use of Web, Cloud, and IoT technologies is compliance with the reasonable requirements of specifications and standards that provide recommendations for minimizing risks and various problems due to failures, cyber-attacks, etc.

Despite the dynamic and successful development of cloud and IoT technologies, their use in domains sensitive to the consequences of failures has certain limitations [4–6].

Additional limitations are cloud regulatory requirements for data privacy [7]. By themselves, these technologies do not minimize the probability of failures and cyber-attacks, and the risks of accidents and disasters caused by them for critical systems and infrastructures. To minimize these risks, first, it is necessary to establish means of ensuring fault and intrusion tolerance, when developing system projects, to implement and maintain such means during the operation of systems, improving them with attention to application experience.

To develop and implement such tools, it is necessary to accurately and adequately assess the relevant indicators of reliability and cybersecurity. This is a difficult task, considering various circumstances [7–10].

- Cloud and IoT systems are complex, multi-component, distributed systems. Therefore, a certain level of generalization is necessary for the analysis and assessment of reliability and cybersecurity, which determines the risks to the accuracy of calculating indicators. Indeed, such systems consist of hundreds and thousands of software and hardware components that have different vulnerabilities and intensities of failures and different laws of distribution of time to failure and recovery, as well as their values. Cloud and IoT systems (CISs) have many common features: they are often combined in integrated cyber-physical systems and IT infrastructure; therefore, it is advisable to solve the problems of assessment and provision of reliability, availability, and cybersecurity from a single point of view.
- The peculiarity of CISs is that their structures and values of parameters affecting the evaluation of indicators are not fixed, although cloud providers providing relevant services use simplified methods based on determining the ratio of the time when the system is in an operational condition until the total time of its use, considering downtime. This is a simplified definition of the stationary value of the availability function, considering downtime for various reasons (failures, cyber-attacks, lack of information sources).
- CISs operate in conditions of uncertainty and changes in the parameters of the physical and informational environment. This factor is particularly influential, as it causes uncertainty in the assessment of dependability and cybersecurity parameters of individual components and systems as a whole.

Thus, there are certain challenges regarding the choice of methods and the types of models for the evaluation of cloud and IoT systems. These challenges are related to:

- firstly, the possibility of the reliable selection and construction of sufficiently fixed models, that is, models selected and intended for the evaluation of systems throughout the entire life cycle, including the operation stage;
- secondly, the possibility of building such sufficiently fixed models in general, considering various factors that lead to changes in the system and environmental parameters.

Therefore, first of all, it is important to analyze various methods and model approaches that are based on the most developed and tested Markov and semi-Markov models [7,10] and that used to assess reliability, availability, and cybersecurity, and secondly, to develop a methodology that will provide an answer to the challenges related to the choice of a model base and its rational change for the monitoring and reliable evaluation of cloud and IoT systems during their use and evolution, taking into account the physical and cyber environment.

The objective of this paper is to improve the accuracy and trustworthiness of cloud and IoT system availability and cybersecurity assessment through informed and flexible choices and the combined use of different types of Markov and semi-Markov models (MMs and SMMs).

The following research questions were formulated to address this objective:

- What approach can be utilized to provide accurate CIS dependability and availability assessments considering cybersecurity issues?

- Which model attributes are important to classify Markov, semi-Markov, and other derived models that are applied to assess CISs and formulate the tasks to choose appropriate models?
- What consequences should be accepted and combined with the different types of MMs and SMMs for CISs considering the features and parameters of the physical and cyber environment?
- How to apply the methodology of choice and combination of MMs, SMMs, and their modifications to assess the availability and cybersecurity of implemented CISs?

The overall contribution of this research covers the developed methodology of decision-making support including a strategy and principles for assessing evolvable cloud and IoT systems' availability/accessibility depending on the reliability, cybersecurity and privacy attributes. They are based on: consideration of continuous evolution of systems and changing CIS parameters; collecting and analyzing data about faults, vulnerabilities and cyberattacks causing failures; actualization of models set, which can be applied for assessment by models' choice and combining taking into account changed conditions and events. Besides, the applied side of the investigation contribution is the algorithms for choice and combining models of dependability assessment in the development, modernization and operation of CIS.

The paper is structured as follows. Section 3 describes the approach to the assessment of CIS dependability and cybersecurity. Section 4 provides results of Markov and semi-Markov models' classification and analysis to choose types of models for cloud and IoT systems assessment. In Section 5, principles and algorithms of choice and combining MMs and SMMs are developed to provide trustworthiness of CIS dependability, availability, and cybersecurity assessment. Section 6 describes and discusses cases related to the choice and combination models to assess real cloud and IoT systems availability and cybersecurity such as cloud video systems, smart building IoT automation systems, unmanned robotics IoT systems, etc. The results of applying the suggested approach to the choice and combining CIS dependability models are analyzed in Section 7. Finally, Section 8 concludes and discusses research results, and suggests an outline of future directions.

## 2. State of the Art

The analyzed references considering Markov modelling for assessing the availability and cybersecurity of cloud and IoT systems can be divided into three groups: the references related to assessing availability and reliability [11–27], the references related to assessing cybersecurity [28–39], and references related to assessing both availability and cybersecurity as attributes of dependability of cloud and IoT systems [40–43].

The authors of [11] considered a few scenarios of virtual machine (VM) practices for single system failure, multiple system failures, and power outages by utilizing a Markov model. This allowed for the investigation of improved repair strategies of accessibility in various circumstances. The availability analysis of the cloud infrastructure [12] is based on a scalable, stochastic model-driven approach to quantify the availability of a large-scale IaaS cloud. In the paper [13], together with reliability block diagrams (RBDs) and stochastic Petri nets, continuous-time Markov chains (CTMCs) were used for developing a hierarchical model-based strategy to evaluate the availability and performance-related metrics for private cloud storage services. The applicability of the proposed models was demonstrated through a cloud storage service hosted on the Eucalyptus platform considering security and privacy issues.

The design of a multi-state semi-Markov availability and reliability prediction model for nodes in volunteer cloud computing systems was discussed in [14]. The implementation of the model in a real volunteer cloud system was presented, and the obtained results were discussed. To analyze the survivability of the cloud service after a service breakdown occurrence, the authors of [15] proposed a model and closed-form solutions involving a CTMC. The model allowed quantitatively assessing the system survivability while

providing insights into the efforts invested in system recovery strategies. Based on the simulation results, certain clarifications of the possible use of the developed models were suggested. In the paper [16], a Markov model was utilized to develop a novel VM migration algorithm aimed at predicting the future load of the host. For evaluating the availability of the algorithm, the cloud simulation software CloudSim was applied. The number of active hosts, the migration times of the VMs, and the energy consumption were considered to show the main advantages of the algorithm compared to other algorithms.

The study [17] addressed a Markov reliability model for analyzing the performance of a multistate cloud computing transition system. The model allowed assessing various reliability measures such as reliability and availability functions, mean time to failure, and mean time to repair. The work [18] proposed models utilizing RBDs and the semi-Markov process to assess the availability of vehicular clouds with a multilayered architecture. These models were developed for each subsystem of the vehicular system, and they were combined for evaluating the availability of the complete system. Authors of [19] presented a Markov process-based reliability model of a flood alerting system (FAS) based on the IoT. The model utilization allowed calculating the reliability and availability functions, as well as the mean time to failure of the FAS and its components. The study [20] presented a modelling methodology based on a hierarchical model of three levels. In addition to an RBD (at the top level), which was utilized for capturing the overall architecture of the IoT infrastructure, and a fault tree (at the middle level), which was utilized for elaborating system architectures of the member systems in the IoT infrastructure, a CTMC (at the bottom level), which was utilized for capturing detailed operative states and transitions of the bottom subsystems in the IoT infrastructure, was considered. A feature of the proposed methodology is the combined assessment of reliability and cybersecurity, which has been demonstrated for IoT smart factory infrastructure. To predict the future availability and reliability of processing requests in a mobile cloud computing (MCC) system, a semi-Markov processes-based multi-state model was defined in [21]. Utilizing this model, decision-making on increasing the speed, disk space of the system, etc. can be supported, and the efficiency of the request processing in the MCC system can be evaluated.

The paper [22] proposed an availability model of a healthcare IoT system comprising two groups of structures described by separate Markov state-space models. These models were combined for modelling the whole IoT system and deriving the probabilities of the full service, the degraded service, and the system unavailability under a given scenario. The paper [23] was devoted to joint utilization of Markov and semi-Markov processes to simulate the behavior of a cloud server system (CSS) and assess its availability considering diverse factors influencing its states. Comparative analysis of simulation results derived via Markov and semi-Markov availability models of the CSS was conducted in the paper. These results can be used by developers and service personnel of the CSS systems to describe the dynamic degradation at different design and functioning phases and consider the impact facets of diverse inception.

The study [24] presented Markov models for the cloud, fog, and edge systems, which allowed defining a range of system metrics to study the performance of workflow scheduling and offloading of service-based applications. An evidence-based stochastic (Markov chains) analysis of the Reactive Architecture with a Cloud accountability System was conducted in [25]. The analysis results were utilized for facilitating a method to provide dependability assurance evidence for the Reactive Architecture. A multistage optimization problem of moving target defense mechanisms deployment was modelled in [26] as constrained by Markov decision processes. This modelling allowed maximizing the available resources of an intelligent system under the limitations of industrial Internet of Things environments. The authors of [27] proposed an auto-scaling algorithm for an elastic cloud workflow engine utilizing reinforcement learning and a semi-Markov decision process (SMDP). The algorithm provides an opportunity to automatically scale instances in advance and adapts to changes in traffic. It was demonstrated that the algorithm made it

possible to reduce the violation rate in Service Level Agreements and improve the availability of the cloud workflow service.

In the paper [28], a Markov model for backhaul link quality was used when developing a novel decentralized authentication architecture aimed at supporting flexible and low-cost local authentication with the awareness of context information of 5G mobile edge computing (MEC) network elements. The simulated results revealed that the architecture using the Markov model could be applied to achieve a flexible balance between network operating cost and MEC reliability. For computing the probability distribution of cloud security threats, a novel approach based on a Markov chain and common vulnerability scoring System was proposed in [29]. The approach allowed estimating the probabilities of cloud threats and types of attacks, which were confirmed by the simulation results. To explore the relationship between physical servers' failure rates and job failure events, a CTMC reliability model for Google cluster physical machines was presented in [30]. The reliability model was utilized for evaluating steady-state availability, steady-state unavailability, mean time to failure, and mean time to repair in the Google cluster.

For optimally and dynamically controlling the assignment of VMs based on the types of service and their stochastic features, cost of security services, cost of blocking, and cost of location, the work [31] presented a method based on an SMDP. This method was used when developing optimal security- and location-aware VM mechanisms to efficiently manage the placement of computation tasks in a multi-cloud environment considering MEC and the backup cloud. In [32], the IP address allocation behaviors in two major cloud computing providers (Amazon Web Services and Google Cloud Platform) were analyzed. A Markov model was utilized for generating an address prediction set from time series data of collected IP addresses. The model allowed reducing the search space for allocated IP addresses. To monitor, detect and localize performance anomalies for container-based clusters, a framework using the hierarchical hidden Markov model was presented in [33]. The results obtained showed that the model could be applied for accurately detecting and localizing performance anomalies in a timely fashion. To realize a secure execution of offloaded tasks in the 5G-driven MEC while minimizing service rejection and security risk, a secure VM management mechanism using the SMDP framework was formulated in [34].

For evaluating the efficiency of the blockchain-based network utilized to increase the security of the network, a discrete-time Markov chain (DTMC) model was built and discussed in [35]. This model allows finding the main influence factors for the efficiency of the network and strategies to improve its efficiency. To determine the steady state availability and the mean time to failure of the cloud under an economic denial of sustainability (EDoS) attack, the work [36] presented a semi-Markov model. This model is a part of the cost management strategy aimed at preventing a cloud adopter from undergoing bankruptcy. To detect intruders in ad hoc mobile cloud computing networks through intelligent cross-layer analysis, a Markov model was developed and explored in [37]. The model was simulated via Network Simulator 3 for evaluating parameters such as accuracy, end-to-end delay, energy consumption, network lifetime, packet delivery ratio, and throughput.

In the study [38], Markov chains in conjunction with the fault trees were used by a hierarchical availability model for evaluating the edge-fog-cloud continuum's availability. The possibility of utilizing the model to support the scalability and capacity planning of edge, fog, and cloud computing environments was demonstrated. To predict and detect the probability of occurrence of security threats and attacks arising in the cloud environment, hidden Markov models were utilized in [39]. The model was trained to identify anomalous sequences or threats by properly detecting accurate and up-to-date information on the risk exposure of cloud-hosted services. The model can be used as an underlying framework and a guiding tool for cloud systems security experts and administrators aiming to secure processes and services over the cloud.

To simulate the behavior of an intrusion tolerance system (ITS) aimed at maintaining a useful level of operational capability throughout ongoing cyber-attacks, a semi-Markov

process was utilized [40]. This also allowed determining the ITS availability, mean time to security failure, and cost to quantitatively analyze the ITS security. The paper [41] presented a case study where, to simulate a cloud-redundant array of an independent disk storage system under denial-of-service attacks, an analytical method integrating Markov chains and binary decision diagrams were used. The case study was presented to demonstrate a risk assessment approach addressing both security and reliability issues. To capture the behavior of a cloud-based firewall service consisting of a load balancer and virtual firewalls, a Markov chain-based analytical model was developed in [42]. From this model, to meet the specified response time, closed-form formulae for determining the minimum number of virtual firewalls were obtained. Numerical examples of the model utilization for achieving proper elasticity under fluctuating traffic load were presented.

The authors of [43] developed a Markov model for simulating a distributed denial-of-service attack based on VM co-residence and analyzing the performance of the cloud data center. To demonstrate the impact of VM co-residence on the performance of physical machines, experiments via the developed model were conducted.

Table 1 shows the possibilities of the models presented in [11–43] for the assessment of cloud/IoT systems.

**Table 1.** Possibilities of the models presented in [11–43] for assessment of cloud/IoT systems.

| Reference | Systems Considered | | Models Utilized | | Characteristics Assessed | | | | |
|---|---|---|---|---|---|---|---|---|---|
| | Cloud | IoT | Markov | Semi-Markov | Availability | Reliability | Cyber Security | Dependability | Intrusion Tolerance |
| [11] | + | − | + | − | + | + | − | − | − |
| [12] | + | − | + | − | + | + | − | − | − |
| [13] | + | − | + | − | + | + | − | − | − |
| [14] | + | − | − | + | + | + | − | − | − |
| [15] | + | − | − | + | + | + | − | − | − |
| [16] | + | − | + | − | + | + | − | − | − |
| [17] | + | − | + | − | + | + | − | − | − |
| [18] | + | − | − | + | + | + | − | − | − |
| [19] | − | + | + | − | + | + | − | − | − |
| [20] | − | + | + | − | + | + | − | − | − |
| [21] | + | − | − | + | + | + | − | − | − |
| [22] | − | + | + | − | + | + | − | − | − |
| [23] | + | − | + | + | + | + | − | − | − |
| [24] | + | − | + | − | + | + | − | − | − |
| [25] | + | − | + | − | + | + | − | − | − |
| [26] | − | + | + | − | + | + | − | − | − |
| [27] | + | − | − | + | + | + | − | − | − |
| [28] | − | + | + | − | − | − | + | − | − |
| [29] | + | − | + | − | − | − | + | − | − |
| [30] | + | − | + | − | − | − | + | − | − |
| [31] | + | − | − | + | − | − | + | − | − |
| [32] | + | − | + | − | − | − | + | − | − |
| [33] | + | − | + | − | − | − | + | − | − |
| [34] | − | + | − | + | − | − | + | − | − |
| [35] | − | + | + | − | − | − | + | − | − |
| [36] | + | − | − | + | − | − | + | − | − |

| | | | | | | | | | |
|---|---|---|---|---|---|---|---|---|---|
| [37] | + | − | + | − | − | − | + | − | − |
| [38] | + | − | + | − | − | − | + | − | − |
| [39] | + | − | + | − | − | − | + | − | − |
| [40] | − | + | − | + | − | − | − | + | + |
| [41] | + | − | + | − | − | − | − | + | + |
| [42] | + | − | + | − | − | − | − | + | + |
| [43] | + | − | + | − | − | − | − | + | + |

The results of the analysis in Table 1 are the following:

- Cloud and IoT systems were considered in 25 and 8 papers, respectively;
- Only Markov and semi-Markov models were utilized in 23 and 9 papers, respectively;
- Both Markov and semi-Markov models were utilized in 1 paper;
- Approaches to availability and reliability assessment of IoT/Cloud systems were presented in 17 papers;
- Twelve papers demonstrated approaches to cybersecurity assessment of IoT/Cloud systems;
- Four papers considered the intrusion tolerance issues related to IoT/Cloud systems.

Thus, there were almost no works devoted to joint utilization of Markov and semi-Markov models for assessing the availability and cybersecurity of cloud and IoT systems.

The general conclusion considering the analysis of the publications concerning the assessment of CIS dependability and cybersecurity using different types of models, especially MMs and SMMs, is the following. Most of them provide a fixed choice of models, without detailed consideration for alternatives and characteristics of individual components, changes in parameters over time and evolution of systems, or physical and cyber environments.

## 3. Approach and Stages

The proposed approach to the assessment of CISs is based on the strategy of model design consisting of the step-by-step selection, adaptation, and possible change of the type and parameters of the models during the use of the system. This strategy is implemented as follows.

### 3.1. Selection of Models

Systems, subject to compliance with the basic assumptions, are evaluated by Markov (MM) and semi-Markov (SMM) models or their modifications, in particular, multi-fragmental (MFM) and multi-phase (MPM) models, which:

- are (not only) selected in such a way that the peculiarities of systems, as well as physical and cyber environments during their development, are considered,
- (but also) vary (change) with the systems during their use, so that various events, changes in failure parameters, cyber-attacks, privacy violations, recovery, etc. are considered for improving the adequacy and accuracy of the assessment of availability and cybersecurity.

### 3.2. Combination of Models

An important component of the approach also offers the possibility to combine models both in time (time combination, t.c.) and by individual components or subsystems (component combination, c.c.).

Therefore, it is about ensuring an adequate selection and "rejuvenation" of the model base of MMs, SMMs, and the use of combined models for the evaluation of cloud and IoT systems. Combining and fitting models in space "time-components" provide increasing adequacy and accuracy of CIS dependability and cybersecurity assessment.

### 3.3. Attributes and Indicators

For evaluation, the following dependability attributes [44,45] were chosen:

- reliability as a continuity of correct functioning and delivery of service;
- availability as availability for correct functioning and delivery of service at any time;
- cybersecurity as a composite of the sub-attributes, first of all, integrity (absence of improper system alterations) and accessibility to services.

Safety (an absence of catastrophic consequences for the user(s) and the environment [44]) is not considered in the framework of the paper. However, the suggested approach can be extended to safety-critical systems as well by the specification of unsafe states and application of safe availability function [45].

The main indicators applied for the assessment of CIS are based on the availability function that considers different reasons for failures including cyber-attacks on system assets. In addition, special indicators of cybersecurity such as rates of attacks on components, the criticality of attacks, and losses of CIS availability caused by cyberattacks are applied.

It should also be noted that in this study, privacy is considered one of the attributes of cybersecurity. To assess privacy using the approach proposed, it is necessary to perform an analysis of cyber-attacks considering the data privacy vector [7,13,46,47].

### 3.4. Stages of Modelling and Assessment

The presented approach is based on the combination of the following main interconnected principles and procedures (Figure 1).

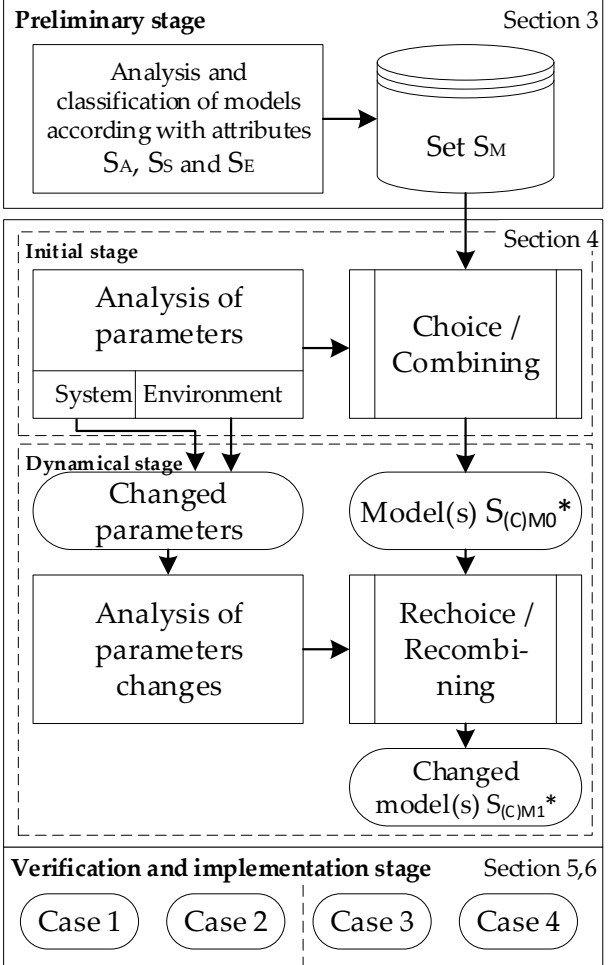

**Figure 1.** Stage of the implementing approach.

1. Preliminary stage: analysis and classification of MMs, SMMs, and their modification according to the specified set of attributes $S_A$ considering different features of systems $S_S$ and factors $S_E$ of the environment. According to the combination of $S_A$, $S_S$, and $S_E$ (in general, their Cartesian product), a set of models $S_M$ is formed:

$$\{S_A \times S_S \times S_E\} \rightarrow S_M, \qquad S_M = \{S_{MM}, S_{SMM}, S_{MFM}, S_{MPM}, S_{CM}\}, \tag{1}$$

where $S_{MM}$, $S_{SMM}$, $S_{MFM}$, $S_{MPM}$, *and* $S_{CM}$ are sets of Markov, semi-Markov, multi-fragmental, multi-phase, and combined models, correspondingly:

$$\begin{aligned}
S_{MM} &= \{M_{M1}, \dots, M_{Ma}\}, \\
S_{SMM} &= \{M_{SM1}, \dots, M_{SMb}\}, \\
S_{MFM} &= \{M_{MF1}, \dots, M_{MFc}\}, \\
S_{MPM} &= \{M_{MP1}, \dots, M_{MPd}\}, \\
S_{CM} &= \{M_{C1}, \dots, M_{Ce}\},
\end{aligned} \tag{2}$$

where *a, b, c, d, e*—numbers of different models for the sets.

A detailed description of the model sets classification and analysis is presented in the next section.

Initial stage: analysis of features of systems $S_S{}^* \in S_S$ and environment $S_E{}^* \in S_E$, and choice of appropriate model $S_{M0}{}^* \in S_M$ or combining models $S_{CM0}{}^* \in S_M$ according to a block of algorithms that will be developed in Section 4.

Dynamical stage: analysis of changing system and environment parameters, different events (failures, cyber-attacks and their effects including violations of privacy, integrity, and confidentiality) and re-selection of the models $S_{M1}{}^* \in S_M$ or combining models $S_{CM1}{}^* \in S_M$ according to developed algorithms (Section 4).

Verification and implementation stage: choice, development and research on the availability and cybersecurity models for several real cloud and IoT systems (Sections 5).

Thus, it is necessary to detail the features of classification of subsets of SMM, SSMM, SMFM, SMPM, and SCM models and suggest a general model classifier. Based on the analysis of publications, and the research and project experience of the authors, the classification of models used or that can be used to evaluate CIS availability and cybersecurity is proposed.

## 4. Classification of the Models for Choice and Combining

A set of models is divided according to the following characteristics:

- the degree of Markovity;
- the number and types of phases;
- the number and types of fragments;
- the possibility of combining models.

### 4.1. The Degree of Markovity

The first attribute is the degree of Markovity (DoM) of the process describing the dependability-related behavior of CIS (Markov and semi-Markov models, the sets $S_{MM}$ and $S_{SMM}$).

### 4.2. Multiphase Models

The next attribute is the number of phases that can be used to describe the behavior of a CIS considering changes in the system parameters and/or states (one and multi-phase models, the sets $S_{OPM}$ and $S_{MPM}$). Multiphase models (MPM) are used to describe time-deterministic events that affect the values of system parameters. Figure 2 shows an example of a multiphase model of an imaginary single channel system with a scheduled maintenance procedure. During the time intervals (phases) between events that change the parameters of the system, its behavior is modelled by a Markov process (a letter within a phase in Figure 2).

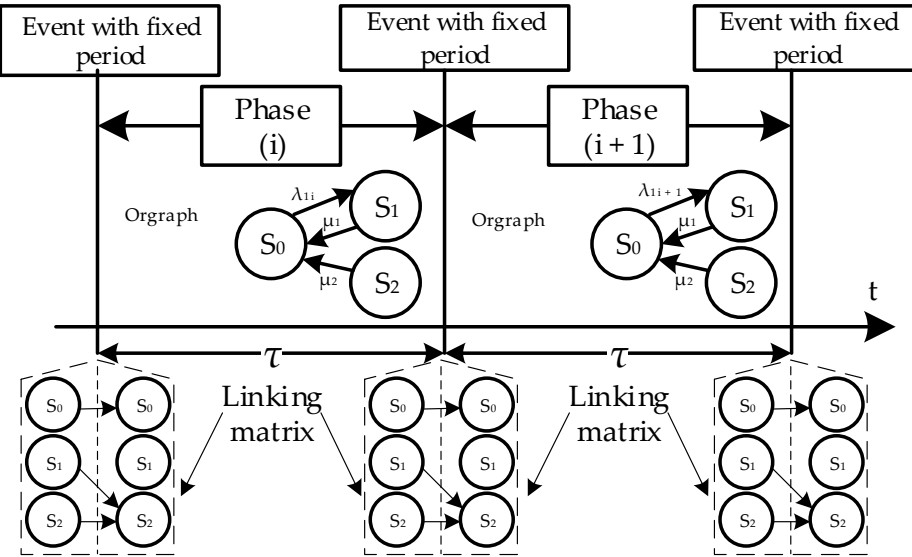

**Figure 2.** Example of a multiphase model.

As an example, a one-component regenerative system is considered, which can be in the operational (S0) and non-operational (S1) states. State S2 simulates maintenance, during which the design defect is eliminated, and as a result, the intensity of failures $\lambda_1$ changes (decreases). The oriented graph does not have transitions to the S2 state, since the service start event, which causes a change in the system parameters, occurs at some point in time (in Figure 2—after a time interval τ). At these moments linking matrix is used. The linking matrix [L] is used to calculate the initial conditions at the beginning of phase (i + 1), based on the probabilities of the conditions at the end of the phase (i), which are mathematically written in the form of Equation (3).

$$\begin{bmatrix} P_0(0) \\ P_1(0) \\ P_2(0) \end{bmatrix} = \begin{bmatrix} 1\ 0\ 0 \\ 0\ 0\ 1 \\ 0\ 0\ 1 \end{bmatrix} \cdot \begin{bmatrix} P_0(\tau) \\ P_1(\tau) \\ P_2(\tau) \end{bmatrix} = \overrightarrow{P_{j+1}}(0) = [L]\overrightarrow{P_j}(\tau) \tag{3}$$

Replacing the probability $P_i(\tau)$ with the values obtained at the previous iteration determines the repetition of Equation (1), which allows one to set the initial conditions at the beginning of each online verification interval (model phase) calculation.

In turn, depending on the **type of period of change**, set $S_{MPM}$ can be divided into models with a fixed value of the period of change…, $S_{MPM,\ f.p.}$ and changed value of the period $S_{MPM,\ ch.p.}$.

### 4.3. Multi-Fragmental Models

The next attribute is **the number of fragments** that describe the behavior of the CIS considering changing parameters of failure rate caused by detection and elimination of design faults and vulnerabilities, recovery rate caused by the increasing complexity of maintenance, etc. According to this attribute, it is distinguished by single and multi-fragmental models (the sets $S_{OFM}$ and $S_{MFM}$).

MFMs are models in which changes in individual parameters and their combinations are described by sets of interconnected fragments of states F1, F2,… If the moments of occurrence of events that cause a change in system parameters are random variables, such events are included in the state space of the oriented graph of the model fragment. Exits from such states initiate a fragment change, and in the next fragment, the system will receive new parameter values (and possibly a new state space).

Figure 3 illustrates a multi-fragment model considering four events related to different types of service procedures. The first fragment contains the initial state of the system S0 and a group of inoperable states S1…S4 caused by maintenance procedures. The second

fragment in Figure 3 contains a subset of states S0*...S4*. The states S0 and S0* in the general macro model are a combination of individual states caused by failures and restoration of system components; therefore, they are marked on the graph with a separate figure—a decagon. State S1 simulates update procedures caused by changing requirements to functionality, security, privacy, etc., S2—vulnerability patching, S3—online verification, and S4—preventive testing. The two-fragment model (Figure 3) describes the removal of one bug/vulnerability after an update or patch. In the case of successive removals of more than one fault, which may lead to CIS failure, the number of fragments of the general model will increase.

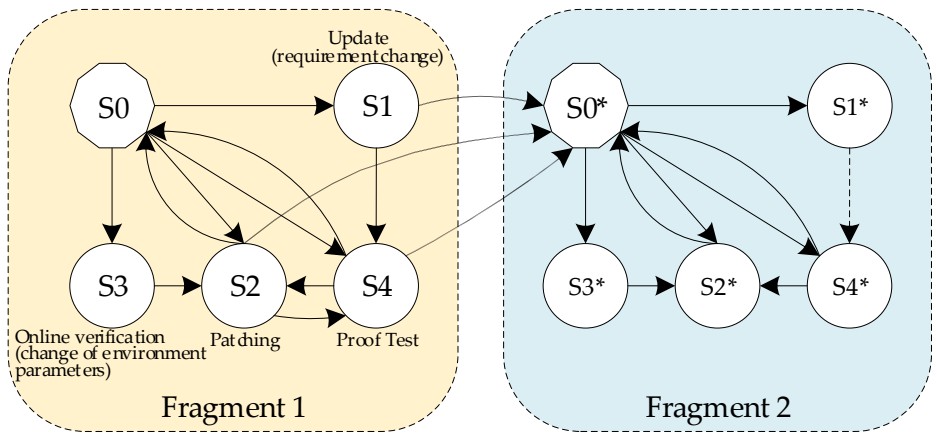

**Figure 3.** An example of a multi-fragmental model.

Besides, set $S_{MFM}$ can be divided into subsets of models according to the subattributes:

- the number of changed parameters: MFMs with one ($S_{MFM, 1\ ch.p.}$) and more ($S_{MFM, m\ ch.p.}$) changed parameters;
- the presence of reverse transitions between fragments: MFMs without reverse transitions ($S_{MFM, r.t.-}$) and MFMs with reverse transitions ($S_{MFM, r.t.+}$). Due to various types of faults and vulnerabilities for some of them, it is advisable to restart without deleting; for others, it is necessary to delete);
- the number of steps of transitions between fragments: MFM with one ($S_{MFM, 1\ s.t.}$) and more ($S_{MFM, n\ s.t.}$) steps of transitions when several faults or vulnerabilities appear and/or are removed at once. In this case, additional distribution laws and their parameters should be identified (for example, for several simultaneously occurring/detected and eliminated design faults and vulnerabilities).

### 4.4. Combined Models

An additional attribute is a possibility of combining different MM and SMM. There are, as mentioned above, two types of combining and subsets of corresponding models: combined model with time ($S_{CM, t.c.}$) and component ($S_{CM,c.c.}$) combining.

In the first case, different models can describe the behavior of the CIS at different operational stages. In the second case, the different models can be applied for the description of different CIS components or subsystems depending on changed conditions.

### 4.5. General Classification

Figure 4 presents a matrix classification of models for CIS dependability assessment. Points mark the presence of relevant attributes/features of the models. Eight models are singled out for sets $S_{MM}$ and $S_{SMM}$:

- one-phase and one-fragmental models $M_{M1}$ and $M_{SM1}$;
- one-phase and multi-fragmental models MM2 and MSM2 with one changed parameter, one-step transition, and without reverse transitions;

- one-phase and multi-fragmental models MM3 and MSM3 with m changed parameters, one-step transition and without reverse transitions;
- one-phase and multi-fragmental models MM4 and MSM4 with one changed parameter, one-step transition, and reverse transitions;
- one-phase and multi-fragmental models MM5 and MSM5 with one changed parameter, n-step transitions and reverse transitions;
- one-phase and multi-fragmental models MM6 and MSM6 with m changed parameters, n-step transition and reverse transitions;
- one-fragmental and multi-phase models MM7 and MSM7 with a fixed value of the period of change;
- one-fragmental and multi-phase models MM8 and MSM8 with the changed value of the period.

There are more complex types of multi-phase models with the multi-fragmental structure of the phase blocks that are not considered in this research.

The set of combined models $S_{CM}$ is a union of the set models of time and component combining $S_{CM} = S_{CM,t.c.} \cup S_{CM,c.c.}$ and is presented by the Cartesian product of the above-described sets: $S_{CM} = \{S_{MM1}, \ldots, S_{MM8}\} \times \{S_{SMM1}, \ldots, S_{SMM8}\}$.

Any combined model of this set can be used for time and component combining. In general, both types of combining can be utilized, i.e., different time combining for different components (subsystems).

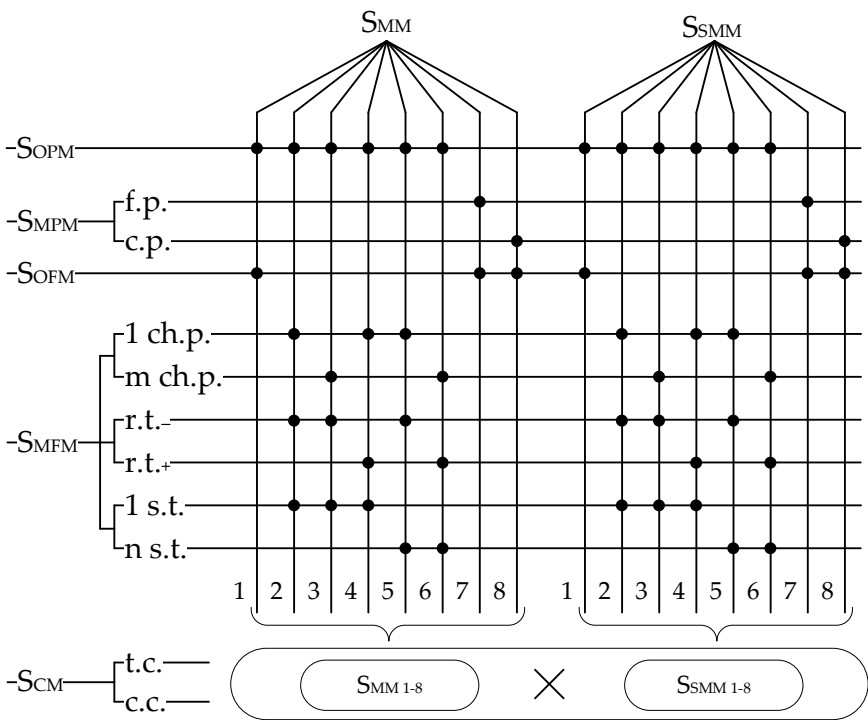

**Figure 4.** Classification of CIS Markov, semi-Markov models.

## 5. Method of Model Choice and Combining

### 5.1. General Algorithm

The method of model selection and combination is illustrated by the diagram of the algorithm in Figure 5. In the initial stage (block 2), the analysis of the operating conditions and system parameters is carried out. The task of this analysis is to determine the possibility of building a single system model (Markov or semi-Markov) (choice block 3). The construction of a single model is possible in the presence of sufficient descriptive statistics

about the states of operation, failure, and repair of system components, and is, as a rule, characteristic of Markov models (blocks 4 and 6).

In some cases, it is advisable to build separate models for each component of the system (for example, for a web server and a database server), which are examined at the same time intervals, with subsequent integration of the obtained results (blocks 5 and 7). Such a case within the proposed concept illustrates components that combine models.

The description of the use of simulation results is illustrated by block 8 (its details are beyond the scope of this article, but, as an example, it is considered in [48]). It is worth noting that for modern CIS, block 8 is constantly used not only in the design stage, but until now, the system will not be disposed of (this is defined by block 12). Blocks 9–11 illustrate the system's response to changes in operating conditions, its architecture, or the parameters of its components. Block 9 must capture and analyze the changes. When the changes are captured (block 10), it must determine whether they have been provided in the system models (models of its components during component combination).

For example, a multi-fragment model can describe a change in the intensity of failures during cyber-attacks, the restoration of a safe state of a component during a proof test in a multiphase model, etc. If there is no change mechanism in the system model, it is necessary to apply the principle of time combination, that is, to return to block 2 and initiate the transition to the initial stage of selection and combination of CIS models.

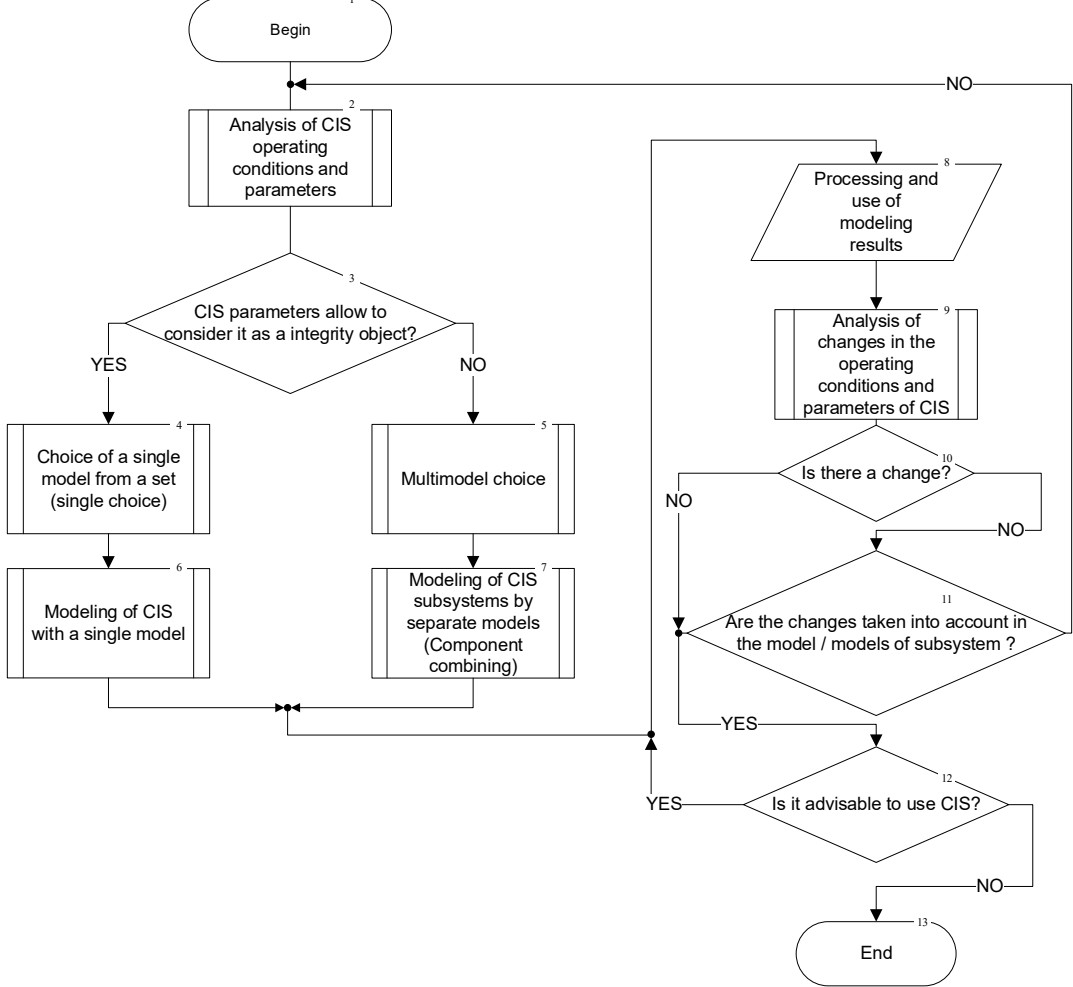

**Figure 5.** The general algorithm of choice and combining models.

In the future, it is worth describing the algorithms of the key procedures of the method, namely blocks 2, 4, and 9.

### 5.2. Algorithm of Choice

The selection of the appropriate model is carried out both for the CIS as a whole (block 4, Figure 5) and for individual components of the system (block 5 in Figure 5). The model selection algorithm is shown in Figure 6 outlined. This algorithm includes the conditions for choosing an adequate mathematical apparatus, considering the conditions of system operation and the principle of minimizing efforts to build a model. For CIS cases with non-renewable components, it is enough to build models of failure trees or RBD (block 4).

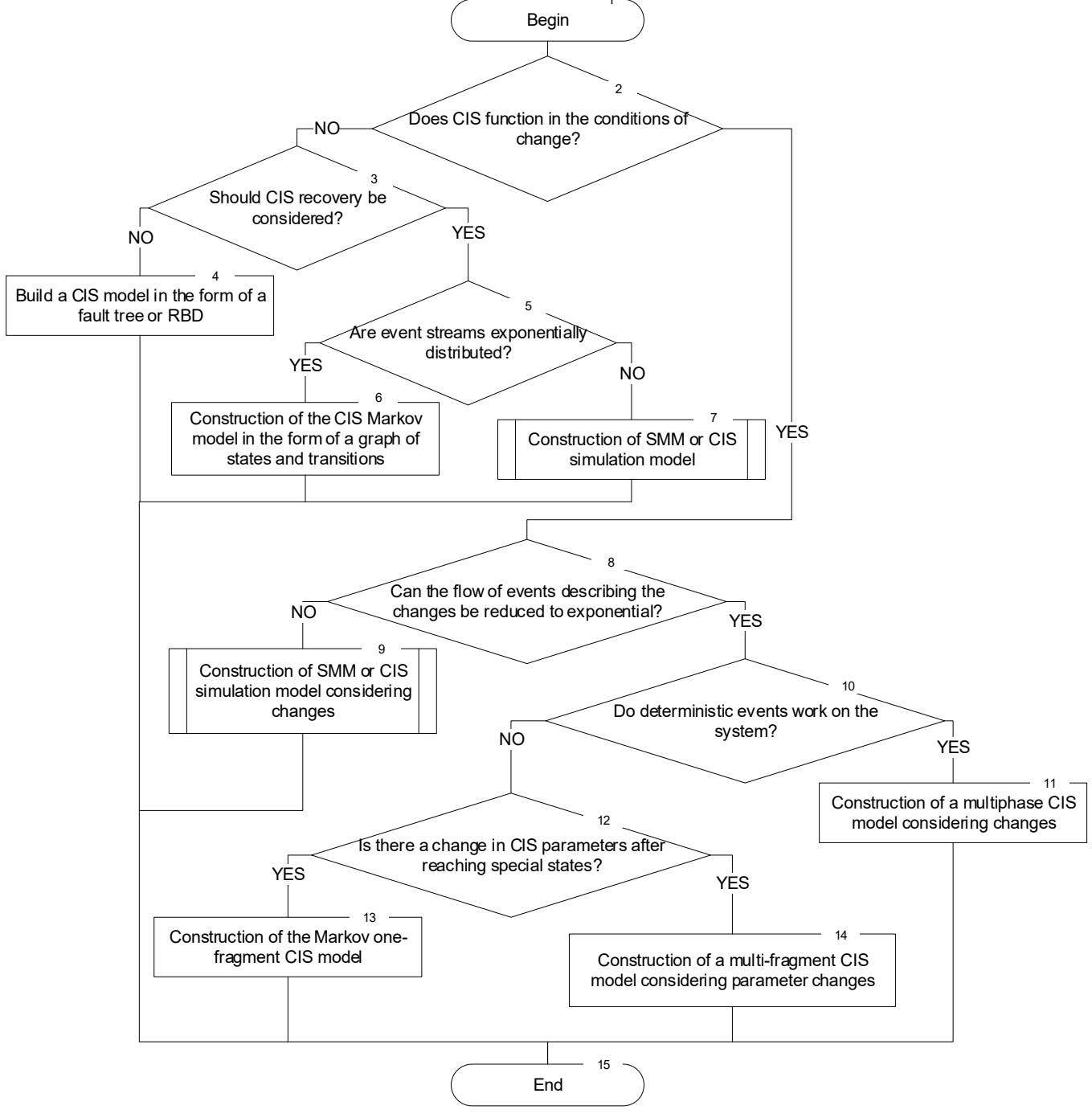

**Figure 6.** Algorithm of model choice.

Block 7 describes the construction of CIS Markov models without considering changes. Blocks 8 and 10 summarize the procedures for selecting and building semi-

Markov and simulation models (as an example, considered in [23,49]). Block 12 describes the construction of a multiphase CIS model. Block 14 describes the construction of single-fragment Markov models with maintenance procedures. Block 15 describes the construction of multi-fragment CIS models considering parameter changes.

*5.3. Algorithms of Data Analysis for Combining Models*

The proposed method includes the procedures of primary and repeated analysis (through operational data processing) and initialization of the time combination principle when the initial CIS model (CIS component models) does not correspond to a change in operating conditions or parameters. The details of the P2 (primary analysis) and P9 (repeated analysis) procedures are shown in Figure 7. Vulnerability repositories, bug tracking, and analytical studies (articles, reviews, monographs, etc.) can be used as data sources for analysis.

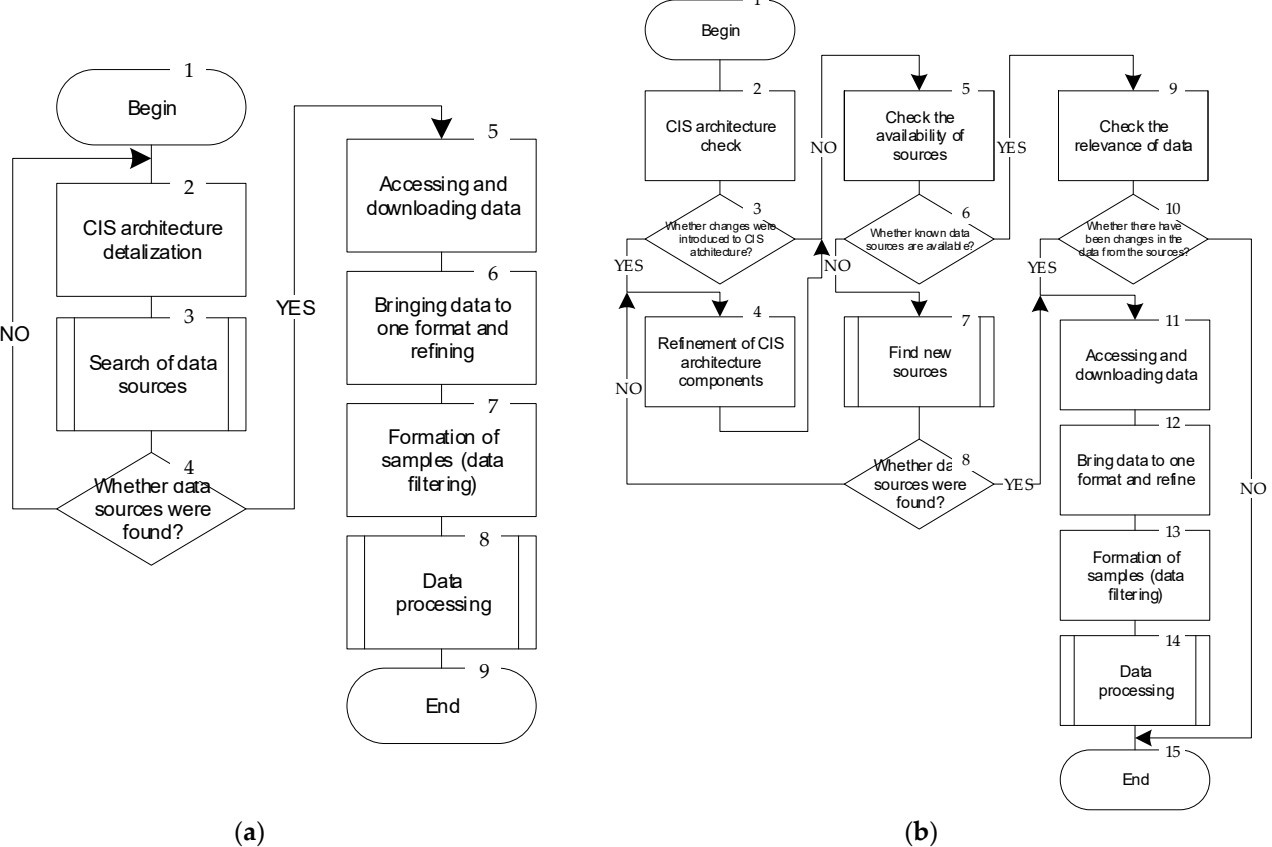

**Figure 7.** Algorithms of primary (**a**) and repeated (**b**) data analysis during combination.

To summarize the considered algorithms, it should be noted that the selection of an adequate CIS model is carried out by an expert method and must consider:

- statistical studies of input parameters of the system, results of checking statistical hypotheses about compliance with the exponential distribution law (or other corresponding distribution laws);
- requirements of control documents, standards, norms, rules, contracts, and customer agreements if they indicate the type or class of the model;
- time and computational resources for modelling (a model that is too detailed may be incalculable due to a lack of resources);
- analysis of the work of other researchers, ease of use of the resulting indicators, and other secondary factors.

**6. Implementation of Suggested Approach and Algorithms of Choice and Combining Models**

Next, cases illustrating the choice and elements of the combination of CIS models will be considered. First, a single model of the four-component cloud server system is proposed; Markov and semi-Markov models are selected for comparison. In the second case, a more complex cloud video system architecture, containing 11 components, is considered. For this architecture, an RBD model that can be used for component combination is given and compared to the CIS Markov model. To consider attacks on one component, a multi-fragment CIS model was built.

In the third case study, a multiphase model is selected, built, and investigated for an unattended robotic IoT system. In the fourth case, the web service of the cloud system model is also considered. To consider the changes in parameters caused by cyber-attacks, a multi-fragment model was chosen, built, and investigated. Other models, in particular, models of failure trees and cyber-attacks on CIS, FMECA/IMECA techniques, as well as their simulation modelling methods, can be found in [50,51].

*6.1. System 1: Cloud Server System*

6.1.1. General Description of the CSS

As indicated in stages 1.2, 2, and 3, the question of modelling cloud systems is currently being worked out in detail, but most studies use the principle of building a single and fixed model. Taking this into account, further attention will be drawn in this case to the issue of choosing an adequate model and rejecting the simulation results in case of a wrong choice.

Since cloud systems have renewable components, block 5 will be used in the selection procedure (Figure 6) (checking the exponential law of the distribution of failure and recovery flow). Next, for comparison, two options will be considered:

- the first option, when block 6 is involved and a Markov model is built;
- the second option, when block 7 of the selection algorithms for building a semi-Markov model is involved.

This case shows a model of a system that functions in perfectly protected conditions against cyber criminals. When considering the semi-Markov model, it is emphasized that the change of the failure intensity parameter with the Weibull distribution can also be used to describe the failures caused by attacks on the system in the presence of supporting statistics. When building CSS models, the "time complexity" principle was not used, since there are no statistics for the modelling system that would predict temporal changes in the parameters of its components.

6.1.2. Development of the CSS Model

The architecture of the cloud system is considered in [23]. The CSS component contains four subsystems, as shown in Figure 8.

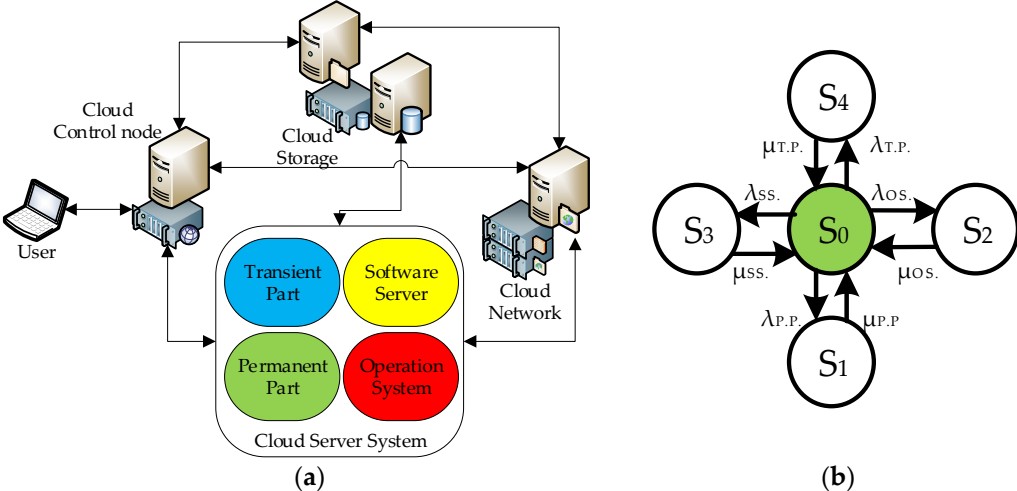

**Figure 8.** The architecture of the cloud computing system (**a**) and Markov graph of its component CSS (**b**).

Markov's Availability Model

The constructed Markov model has one working and four down states S = {0,1,2,3,4}. In S0, the CSS system is operational (available). In state S1, the system is unavailable due to permanent part (PP) failure; in state S2, it is unavailable due to an operating system (OS) failure; in state S3, unavailable due to software server (SS) failure; and in state S4, unavailable due to transient part (TP) failure.

Further development of the Markov model is the construction and solution of the system of Kolmogorov–Chapman differential equations. A simplified version of the solution consists of reducing the system to a linear form due to the equality:

$$\sum_{i \in \{0 \ldots 4\}} \lambda_{ij} \cdot p_i = 0, \ j \in \{0 \ldots 4\}. \tag{4}$$

Solving the system of linear equations makes it possible to investigate the stationary availability coefficient of the CSS.

Semi-Markov Availability Model

The oriented graph (Figure 8b) is used to build a semi-Markov CSS model. In [23], two possible options for working out statistical hypotheses regarding the distribution of input parameters are defined. In the first case, the cumulative distribution law (CDF) of the recovery time of the CSS component is distributed according to the Erlang law of the second order:

$$Q_{ij}(t) = 1 - \left(1 + \mu_j t\right)e^{-\mu_j t} = Erlang\left(2, \mu_j\right), \tag{5}$$

where $\mu_j$—repair rate of the CSS components;

$t$—recovery time of the CSS component, random parameter.

In the second case, the law of distribution of the failure flow of the software server component is defined as Weibull:

$$Q_{03}(t) = 1 - e^{-\left(\frac{t}{\beta_3}\right)^{\alpha_3}} = Weibull(\alpha_3, \beta_3), \tag{6}$$

where $\lambda_3 = \frac{1}{\beta_3}$—a failure rate of the software server.

The remaining transition probabilities (as in the Markov model) remain distributed according to the exponential law.

The CSS semi-Markov model includes a matrix of transition probabilities:

$$P = [p_{ij}] = \begin{bmatrix} 0 & p_{01} & p_{02} & p_{03} & p_{04} \\ 1 & 0 & 0 & 0 & 0 \\ 1 & 0 & 0 & 0 & 0 \\ 1 & 0 & 0 & 0 & 0 \\ 1 & 0 & 0 & 0 & 0 \end{bmatrix}. \tag{7}$$

The transient probabilities for Matrix (7) are described as

$$p_{ij} = P_{ij}(t) = \int_0^t \prod_{l \neq j} \left(1 - Q_{il}(u)\right) dQ_{ij}(u). \tag{8}$$

Sojourn time distribution $H_i(t)$ and mean sojourn time $h_i$ at state $i$ have been determined according to the main provisions for the SMMPs [51]. As a result, the steady-state availability of CSS for relevant vector $\pi = \{\pi_0, \pi_1, \pi_2, \pi_3, \pi_4\}$ is described by the expression:

$$A_{CSS} = \frac{h_0}{h_0 + 2\left(\frac{p_{01}}{\mu_1} + \frac{p_{02}}{\mu_2} + \frac{p_{03}}{\mu_3} + \frac{p_{04}}{\mu_4}\right)}. \tag{9}$$

6.1.3. Research and Analysis of the Results

The numerical data of the parameters for the relevant familiar availability model [52] are listed in Table 2.

**Table 2.** Numerical Values of Modeling Parameters.

| Parameters | Value (1/hour) | Parameters | Value (1/Hour) |
|------------|----------------|------------|----------------|
| $\lambda_{P.P}$ | 0.0014 | $\mu_{P.P}$ | 0.1667 |
| $\lambda_{OS}$ | 0.0042 | $\mu_{OS}$ | 12 |
| $\lambda_{S.S}$ | 0…0.01 | $\mu_{S.S}$ | 20…70 |
| $\lambda_{T.P}$ | 0.0028 | $\mu_{T.P}$ | 30 |

Figure 9a shows availability dependency and comparative availability assessments based on MMPs and SMMPs for the first situation considering failures of the software server of the CSS system. Figure 9b illustrates comparative availability assessments of the MMPs and SMMPs for the second situation. The modelling results testify that availability assessments based on MMPs and SMMPs are associated and can be employed by developers of the CSSs to create effective functioning systems considering dependability facets.

The following conclusions are based on the simulation results:

- the use of the Markov model allows obtaining the average result of the availability in comparison with the semi-Markov model for different distribution laws;
- if the statistical hypotheses for the input parameters assessment tend to the Weibull distribution with the parameter $\alpha = 0.5…1.5$, using the MM is possible to simplify the modelling. However, such values of the Weibull parameter are rarely applicable for the assessment of cyber-attacks on the CIS, leading to the construction of more complex models.

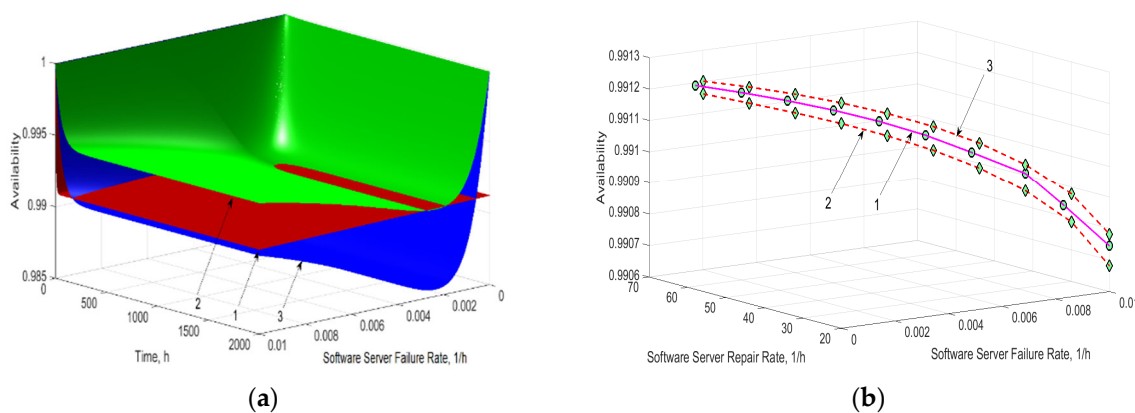

(**a**)                                                        (**b**)

**Figure 9.** Comparative availability assessments for MMPs and SMMPs: (**a**) 1—for MMPs; 2—for SMMPs with CDF Erlang (18, $\mu_j$); 3—for SMMPs with CDF Erlang (11, $\mu_j$). (**b**) 1—for MMPs; 2—for SMMPs with CDF Weibull (0.5, $\beta_3$); 3—for SMMPs with CDF Weibull (1.5, $\beta_3$)

*6.2. System 2: Cloud Video System*

6.2.1. General Description of the CVS

In this example, the construction, comparison and choice of one of the three models are considered:

- RBD for cases of component combination (block 5 of the selection and combination algorithm in Figure 5 is involved);
- multi-fragment model (block 14 of the algorithm in Figure 6);
- for the case when the RBD model combines the Markov models of the different CVS components.

The aspect of cybersecurity is considered when building a multi-fragment model since the change in model fragments is due to a successful attack on the CDN component.

6.2.2. Development of the CVS Model

In this case, an example of the functioning of cloud services for processing video traffic is considered [10]. The video service cloud architecture model contains three levels of virtual networks (mobile, CDN, and primary virtual network) that serve groups of end devices. The CDN is separated from the primary virtual network by the SignalR service and a VPN gateway. Application services (app service API, calls and autoscale service) are located in the primary virtual network. It also hosts the Message Queuing Service (QS) and the Load Balancer (LB). The common elements for a typical cloud system are, therefore, a group of end devices (DSM), physical access network (MNT), virtual access network elements (VPN and SGR) and load balancing (LB). Cloud application services (API, calls, autoscaling) are specialized video hosting services. Such an important element as a CDN should be singled out since the use of a network enables partial offloading of regional cloud services. A CDN also protects against DDoS-type cyber-attacks, but this element is most often attacked among other components.

RBD Availability Model

The reliability block diagram (RBD) of the cloud system (Figure 10) will include seven consecutive elements. Each of the elements characterizes the serviceability of the corresponding elements of the architecture.

Parallel links in RBD create WiFi and MNT, SGR and VPN. Elements are: web servers (LB, QS) and application servers (APS) that can be reserved through a high availability cluster, in which case RBD will contain additional redundant components.

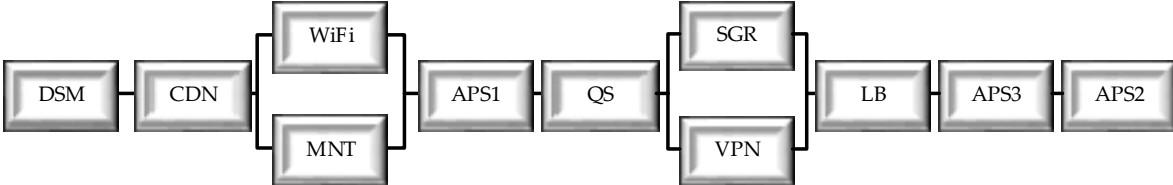

**Figure 10.** Case 2: RBD for CVS.

According to [51], the calculation of the availability of the system with a mixed connection of elements is performed by formula (10):

$$A_{CVS} = A_{DSM} \times A_{CDN} \times A_{WM} \times A_{APS1} \times A_{QS} \times A_{LB} \times A_{APS3} \times A_{APS2}, \tag{10}$$

where

$$A_{WM} = 1 - (1 - A_{WiFi})(1 - A_{MNT}), A_{VS} = 1 - (1 - A_{VPN})(1 - A_{SGR}). \tag{11}$$

The availability values obtained by Formulas (10) and (11) are stationary. This greatly simplifies the model but does not allow studying the dynamics of changes in availability function over time. The availability of each of the components of the cloud architecture in Equation (10) is determined by the formula $\mu_i/(\mu_i+\lambda_i)$, where the values of $\mu_i$ and $\lambda_i$ are the input parameters averaged by the method in [51,52] for each element of the cloud system.

Multi-Fragment Availability Model Considering Attacks on CDN

The multi-fragment model of cloud system availability allows considering the change of input parameters in one model step. This complicates the marked-oriented graph of the functioning of the system, as shown in Figure 11. The process of functioning of the cloud system is as follows. Initially, the system implements all planned functions and is in state S1. In the process of functioning, the failures of the system components are manifested, as a result of which it passes into the states S2…S14 and is restored (the system returns to state S1).

To simplify the perception of the model, in the oriented graph (Figure 11) all transitions not related to the attack on the CDN are hidden in the superstates S (1..14*) (for the first fragment) and S (15..28*) (for the second fragment).

After a certain time interval, the system fails due to an attack on the vulnerability of the CDN component, and it goes into state S3. If the attacker succeeds (the CDN attack was successful), the system moves to a new part of the model (state S17), and if the attack fails, it returns to state S1. The probability of successful attack is weighted by the parameter $a \in [0…1]$. After several successful attacks (usually $Nf = [8…12]$, the attack rate reaches maximum $\lambda_{CDN}^{max}$ (because for technical reasons, the attacker cannot speed them up).

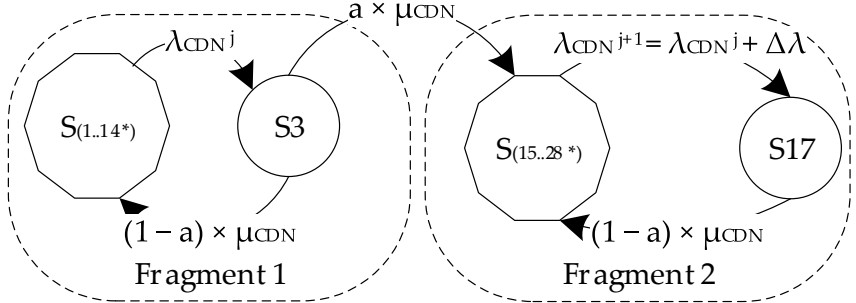

**Figure 11.** Oriented graph of multi-fragment availability models CVS.

The value of the resulting availability indicator in the multi-fragment model is determined by Formula (12).

$$A(t) = \sum_{i=0}^{Nf-1} [P_{14i+1}(t) + P_{14i+4}(t) + P_{14i+6}(t) + P_{14i+10}(t) + P_{14i+12}(t)]. \qquad (12)$$

6.2.3. Research and Analysis of the Results

The primary input parameters of RBD, Markov, and multi-fragment models were determined on the basis of research and certification data [53,54] for the analogue versions of CVS samples. Parameter values are presented in Table 3.

**Table 3.** Constant values of simulation processing input parameters for RBD and Markov models.

| No. | Name of Systems Component | Failure Rate | Value (1/h) | Repair Rate | Value (1/h) |
|-----|---------------------------|--------------|-------------|-------------|-------------|
| 1 | Desktop and Mobile (DSN) | $\lambda dsm$ | 0.000925926 | $\mu dsm$ | 0.02083 |
| 2 | Content Delivery Network Service (CDN) | $\lambda cdn$ | 0.001388889 | $\mu cdn$ | 1 |
| 3 | Wi-Fi | $\lambda wifi$ | 0.001488095 | $\mu wifi$ | 0.04167 |
| 4 | Mobile Network (MNT) | $\lambda mnt$ | 0.000462963 | $\mu mnt$ | 0.5 |
| 5 | App Service (API) | $\lambda aps1$ | 0.002083333 | $\mu aps1$ | 1.5 |
| 6 | Queue Service (QS) | $\lambda qs$ | 0.001302083 | $\mu qs$ | 1 |
| 7 | Load Balancer (LB) | $\lambda lb$ | 0.001190476 | $\mu lb$ | 1 |
| 8 | SignalR Socket Service (SGR) | $\lambda sgr$ | 0.001666667 | $\mu sgr$ | 1 |
| 9 | VPN Gateway | $\lambda vpn$ | 0.001736111 | $\mu vpn$ | 1 |
| 10 | App Service (Calls) | $\lambda aps2$ | 0.00245098 | $\mu aps2$ | 0.66667 |
| 11 | Autoscaling Service | $\lambda aps3$ | 0.002777778 | $\mu aps3$ | 1 |

To investigate the system availability, the variable input parameter $\lambda_{CDNj}$ was taken, reasoned in [54] and summarized in Table 4.

A comparison of RBD and Markov models is performed under the condition $t \to \infty$ (for stable availability). Under this condition, the solution is presented by a system of linear algebraic equations. The results of the calculations are shown in Table 5.

**Table 4.** Variable values of simulation processing input parameters for the multi-fragment model.

| No. | Parameter | Variable | Value (1/h) |
|-----|-----------|----------|-------------|
| 1 | The minimum value of CDN failure rate due to hacker attack | $\lambda cdn\_min$ | 0.001388889 |
| 2 | The maximum value of CDN failure rate due to hacker attack | $\lambda cdn\_max$ | 0.041666667 |
| 3 | Delta of change in CDN failure rate | $\Delta\lambda cdn$ | 0.004027778 |
| 4 | Probability of successful attack | $\alpha$ | 0…1 |
| 5 | Number of fragments in multi-fragment model | nf | 10 |

**Table 5.** Comparison of results of RBD and Markov models.

| No. | Element of System | Markov Model $P_i$ | RBD Model $A_i$ | $\Delta$ $\|P_i - (1 - A_i)\|$ |
|-----|-------------------|--------------------|-----------------|-------------------------------|
| 1 | - | 0.895362266 | - | - |
| 2 | dsm | 0.039793882 | 0.957446805 | 0.002759313 |
| 3 | cdn | 0.001243559 | 0.998613037 | 0.000143404 |
| 4 | wifi | 0.031977218 | 0.965517247 | 0.002505535 |
| 5 | wifi/mnt | $6.66 \times 10^{-4}$ | 0.99929627 | $3.75376 \times 10^{-5}$ |
| 6 | mnt | 0.018653381 | 0.979591837 | 0.001754783 |

| 7 | aps1 | 0.001243559 | 0.998613038 | 0.000143404 |
| 8 | qs | 0.001165836 | 0.99869961 | 0.000134554 |
| 9 | lb | $1.07 \times 10^{-3}$ | 0.99881094 | 0.000123153 |
| 10 | sgr | 0.001492271 | 0.998336106 | 0.000171623 |
| 11 | sgr/vpn | $2.59 \times 10^{-6}$ | 0.999997116 | $2.9295 \times 10^{-07}$ |
| 12 | vpn | 0.001554448 | 0.998266898 | 0.000178654 |
| 13 | aps2 | 0.003291773 | 0.996336997 | 0.000371231 |
| 14 | aps3 | 0.002487118 | 0.997229917 | 0.000282966 |
| $A_{CVS}$ | CVS | 0.949039584 | 0.945631343 | 0.003408241 |

The simulation results showed that the difference between the cloud system availability indicators determined by RBD and Markov models have differences not exceeding $\Delta A = 0.0034$. The weakest elements in the architecture in the absence of attacks are end-user devices (DSM). Figure 12a illustrates the decrease in availability with increasing input parameter $\lambda_{CDNj}$ within the interval of Table 3.

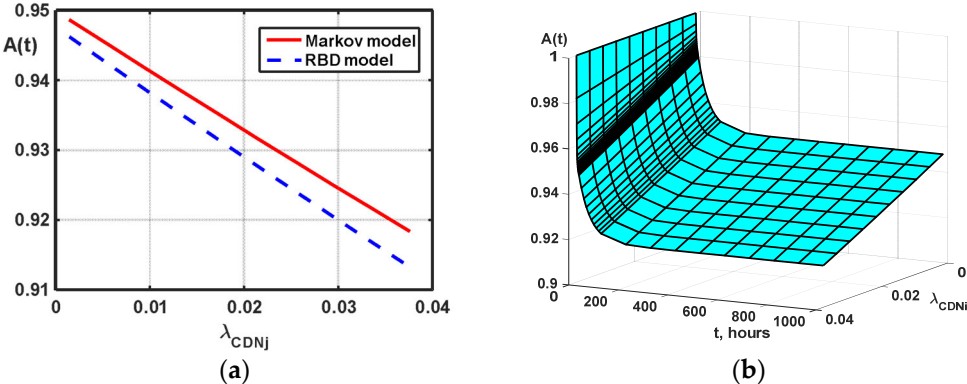

**Figure 12.** Results of RBD-based and Markov models (**a**) and Markov model (**b**) for different value $\lambda_{CDN}$.

Estimates obtained using RBD and MMs with increasing CDN failure rates increase the discrepancy from $\Delta A = 0.0034$ to $\Delta A = 0.0051$.

To solve systems of differential equations constructed according to the Kolmogorov–Chapman matrix, in the paper we used the ode15s function [55]. The simulation results are shown in Figure 12b. The change in time of the availability indicator, illustrated by the Markov model, shows the asymptotic direction of the function to a stationary value during the first t = 200 h of CVS operation.

The results of availability modelling using a multi-fragment model are illustrated in Figure 13. Graphs in Figure 13a allow comparing the results of Markov and multi-fragment models. The availability function obtained by the MFM method reaches a stationary value after t = 8000 h of operation, and the specified value of stationary availability is A = 0.9189. This indicator can be determined by a simpler Markov model, taking the value of the input parameter $\lambda_{CDN}$ equal to $\lambda_{CDN}^{max}$.

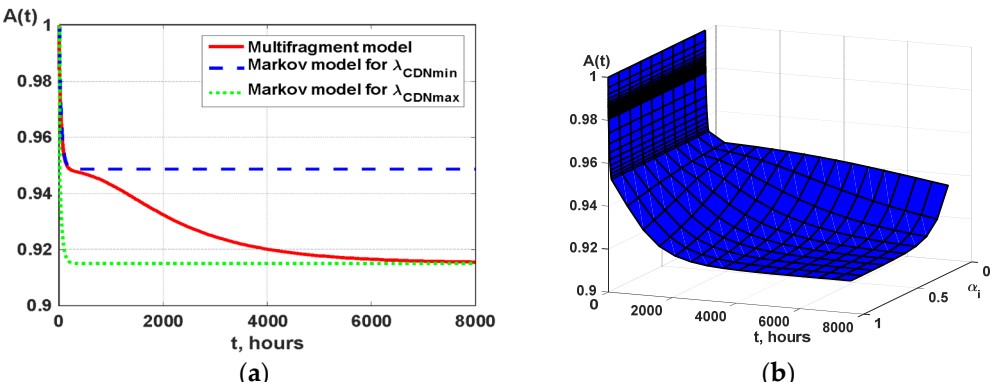

**Figure 13.** Results of Markov and multi-fragment models (**a**), and multi-fragment model (**b**) for different value $\alpha$.

Figure 13b illustrates the influence of the input parameter $\alpha$ on the dynamics of changes in the availability function. The mechanism of influence of this parameter is as follows. As the parameter $\alpha$ increases, the time of transition of the availability function to stationary mode decreases, so for $\alpha = 0.5$, t = 8000 h; for $\alpha = 0.9$, t = 4000 h (that is twice as fast).

### 6.3. System 3: Unmanned Robotic IoT System (URIS)

6.3.1. General Description of URIS

This case illustrates the operation of block 11 of the selection algorithm in Figure 5. The use of the multiphase simulation device is associated with subsystems of the robot system, which work in the mode of low intensity of requests for execution and undergo online verification of the software only at certain times.

For such systems, installation of software patches is only possible when software design errors or vulnerabilities are detected after online verification. The time points of the start of online verifications have a periodic schedule. When assessing cybersecurity, the multiphase model allows you to describe the process of installing security updates, the release of which (unlike patches) has a regular nature.

6.3.2. Development of the URIS Model

The URIS' RBD is shown in Figure 14a.

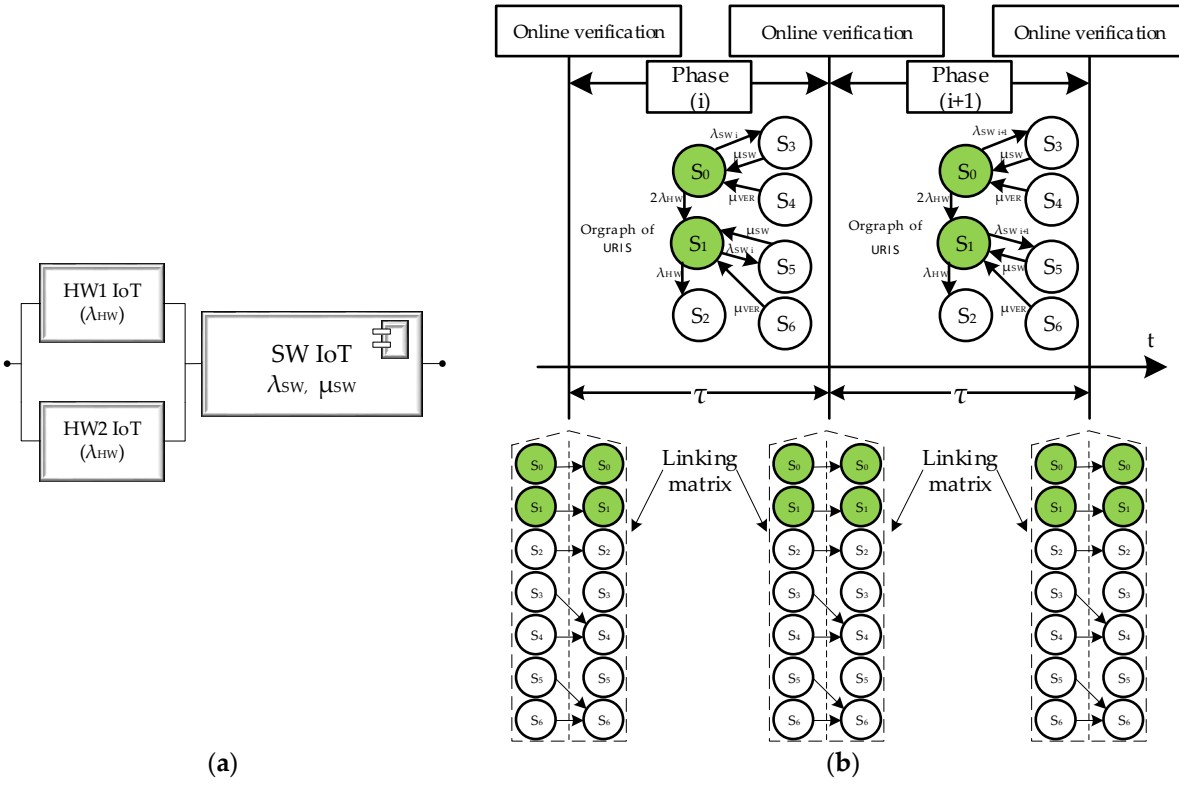

(**a**)　　　　　　　　　　　　　　　　(**b**)

**Figure 14.** RBD (**a**) and multiphase model (**b**) of URIS.

The labelled graph simulating the behavior of the system during the period of one phase (Figure 14b) has 7 states:

- operational (S0 and S1—with a detected failure of one hardware channel),
- inoperable (S2—with two detected hardware errors, S3 and S5—with detected software errors; S4 and S6—states of software patching).

In the time intervals between the online verification of the subsystems of the robotic IoT system, its behavior is modelled by the Markov process in the upper part of Figure 14b: the IoT system can fail due to hardware failures (transitions S0 → S1 and S1 → S2), software failure is modelled by transitions S0 → S3 and S1 → S4, and software restart is modelled as state change S3 → S0 and S4 → S1.

If the IoT system goes into the patch installation states (S4 or S6) at the time of the previous online verification procedure, its successful completion is simulated by changing the states S4 → S0 and S6 → S1. Since verification and patching cannot be started within the time interval of the duration of one phase, the transitions S3 → S4 and S5 → S6 are not possible. Since the online verification procedures revealed the manifestation of DP before entering the S4 and S6 states, the transitions S4 → S0 and S6 → S1 are weighted by the μVER parameter (the intensity of the verification and patching procedures).

Online verification procedures are carried out at certain times (in Figure 14b—the moments to activate the linking matrix (13)). The logic to construct the transitions of this matrix is as follows. If a software error appears during the previous phase, the patching procedure is started (transitions S3 → S4 and S5 → S6). If the component of the robotic IoT system was operational in the previous phase, it remains in the corresponding operational state (S0 → S0, S1 → S1); if the system enters an inoperable state, it remains in it (S2 → S2).

$$\begin{bmatrix} P_0(0) \\ P_1(0) \\ P_2(0) \\ P_3(0) \\ P_4(0) \\ P_5(0) \\ P_6(0) \end{bmatrix}_{i+1} = \begin{bmatrix} 1000000 \\ 0100000 \\ 0010000 \\ 0000100 \\ 0000100 \\ 0000001 \\ 0000001 \end{bmatrix} \cdot \begin{bmatrix} P_0(\tau) \\ P_1(\tau) \\ P_2(\tau) \\ P_3(\tau) \\ P_4(\tau) \\ P_5(\tau) \\ P_6(\tau) \end{bmatrix} = \overrightarrow{P_{i+1}}(0) = [L]\overrightarrow{P_i}(\tau) \tag{13}$$

In the hypothetical case, if the patch started during the previous verification is not completed, the IC remains in the patch state (S4 → S4, S6 → S6). The linking matrix [L] is used to calculate the initial conditions at the beginning of phase (i + 1) based on the probabilities of the conditions at the end of phase (i), which are written mathematically in the form of Equation (13).

The replacement of $P_i(\tau)$ with the values obtained in the previous iteration determines the repetition of equation (13), which makes it possible to calculate the initial conditions at the beginning of each online verification interval (model phase).

The average unavailability index Uavg is calculated by the method described in [51], using the following definite integral

$$U_{avg}(\tau) = \int_0^\tau U(t)dt, \text{ where } U(t) = 1 - P_1(t) - P_2(t). \tag{14}$$

6.3.3. Research and Analysis of the Results

The values of the input parameters of the robotic IoT system model are averaged based on statistical data on the operation of such systems [56]. Fixed values of input parameters are described in Table 6.

**Table 6.** Values of input parameters.

| Symbol | Illustration | Value | Unit |
|--------|-------------|-------|------|
| $\lambda_{HW}$ | failure rate of single hardware channel | $3 \times 10^{-7}$ | 1/hour |
| $\lambda_{SW\,0}$ | the initial software failure rate | $5 \times 10^{-3}$ | 1/hour |
| $\mu_{SW}$ | system recovery rate after the occurrence of software fault | 0.2 | 1/hour |
| $\lambda_{SW\,k}$ | software failure rate after fixing all faults is zero | 0 | 1/hour |
| $\mu_{ver}$ | rate matches the average time verification procedure | 0.0667 | 1/hour |
| $\Delta\lambda_{SW}$ | additional limitation presents elimination of 10 undetected software defects and uniformity of load for their localization and elimination | $\lambda_{SW\,0}/10$ | 1/hour |
| T | The time interval for analysis of availability function behavior | 90,000 | hours |

When building the model, it is necessary to consider the change in the $\lambda_{SW}$ parameter when software errors are eliminated, which occurs after their manifestation and the arrival of the next verification time. However, this event is probable, and it is not possible to say exactly at which time interval $\Delta\lambda_{SW}$ will decrease with $\Delta\lambda_{SW}$. The following technique was used when building a multiphase model of a robotic IoT system. At the start of a new phase, the probability of the manifestation of a software error is defined as the sum of the probabilities $P_4(\tau) + P_6(\tau)$ of the previous phase. Then, the change in the intensity of software errors in the new phase is determined by the formula

$$\Delta\lambda_{SW}(\tau + 1) = \Delta\lambda_{SW} \cdot [P_4(\tau) + P_6(\tau)]. \tag{15}$$

The task of model research is reduced to finding the value of $\Delta T_{VER}$—the period of online verifications, during which the requirement to ensure that the system availability function is not lower than 0.98 for 2 years (20,000 h) of operation.

The view of the curves of the function of instantaneous unavailability ("saw") and average unavailability, obtained with the help of the model of the robotic IoT system, is shown in Figure 15.

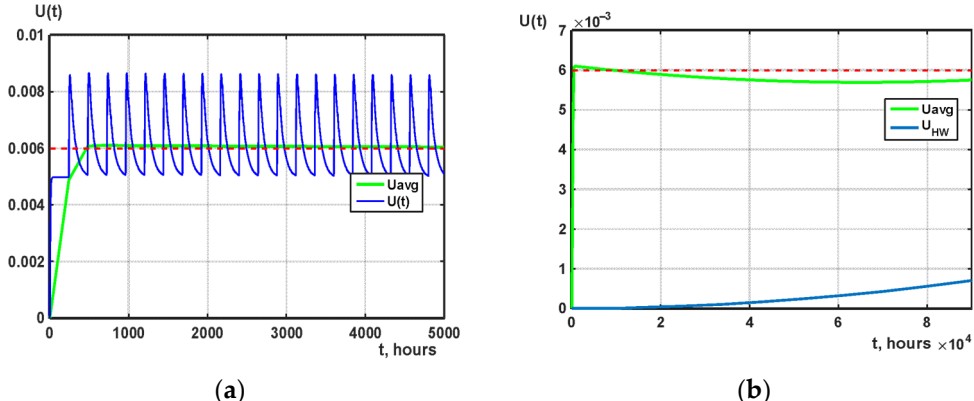

**Figure 15.** Unavailability functions of a robotic IoT system with periodic online software verification obtained using a multiphase model with time intervals (**a**) [0…5000] hours and (**b**) [0…90,000] hours.

On the graph in Figure 15a, the scale of the time interval t = [0…5000] is reduced so that the characteristic "saws" of the instantaneous failure curve are visible. In Figure 15b it is observed that the Uavg curve decreases its values over time. This illustrates the process of eliminating software defects.

The selection method determined the value of the parameter $\Delta T_{VER}$ = 7 days (168 h), which ensures that the value of the system unavailability function is not more than 0.02 for 2 years (20,000 h) of operation (Figure 15b).

Thus, a feature of the developed multiphase model is the consideration of the periodic component of maintenance procedures together with the random component of failure and repair events of SW and HW. This makes it possible to increase the adequate estimate of the indicator of the average unavailability function and to determine the interval of the execution of online verification procedures to ensure an average level of unavailability not lower than 0.02. Considering cyber-attacks will increase the level of unpreparedness of the system compared to the achieved result. From the point of view of the principle of operation of the multiphase model, to counter cyber-attacks, more frequent security updates are required. However, there is a limit—the update period, after which the unavailability of the system will increase due to downtime, which is not caused by attacks, but by performing only security update procedures.

*6.4. System 4: Web Service of Cloud System (WSC)*

6.4.1. General Description of the WSC

In this case, block 14 of the model selection algorithm (Figure 5) is used under the condition of component combination, that is, a multi-fragment model is not built from a complete cloud system, but from its component, a web service.

The aspect of cybersecurity is considered when building a multi-fragment model since the change in model fragments is due to a successful attack on the DNA component.

6.4.2. Development of the WSC Model

A model of a corporate web service was developed to simulate the patch installation processes. The reliability block diagram (RBD) of the corporate web service (Figure 16) includes four consecutive elements, each of which characterizes the health of the domain name service (DNS), the web server (HTTP server), the application service (App), and the database server (BD). In the developed model, it was decided to limit ourselves to the

description of software errors caused by design defects and attacks on component vulnerabilities.

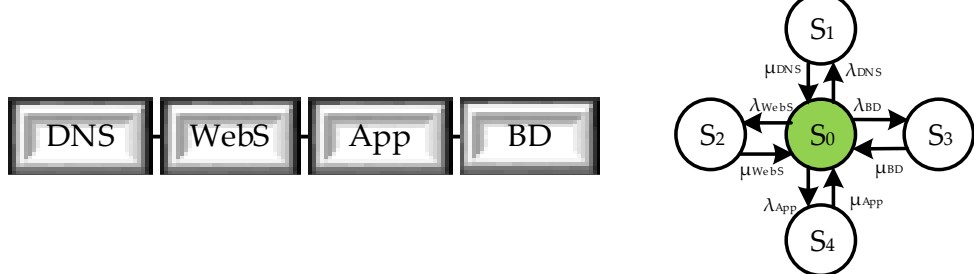

**Figure 16.** RBD of web service support services and labeled graph of the operation of web service support services (one-fragment model).

The developed multi-fragment model of the WCS describes the preventive measures of the security audit to detect and eliminate vulnerabilities and allows the elimination of the vulnerability detected during the attack. The labeled graph of the multi-fragment model for a system with two vulnerabilities is shown in Figure 17.

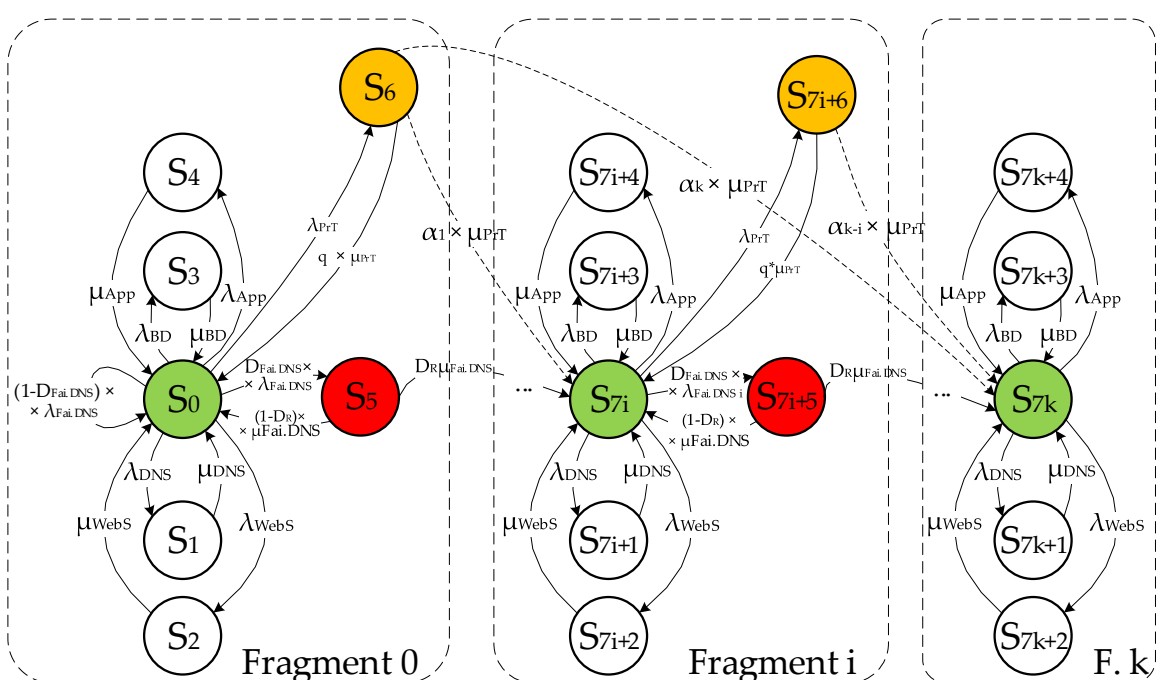

**Figure 17.** The labeled graph of the multi-fragment model of the web service.

According to the graph in Figure 17, initially, the web service operates in states of failure and recovery of DNS services, web server HTTP requests, application servers and database. After an attack on the DNS service (which enters the S5 state with intensity $D_{Fai\,DNS} \times \lambda_{Fai\,DNS}$), the system becomes inoperable, but can recover by restarting without remediation with conditional intensity $(1 - D_R) \times \mu_{Fai\,DNS}$, or with conditional intensity remediation by $D_R \times \mu_{Fai\,DNS}$ intensity. With the intensity $\lambda_{PrT}$, preventive measures (state S6) are performed in the system, whereby 0 to nv vulnerabilities can be detected and eliminated. After the manifestation and elimination of all vulnerabilities, the system continues to function under conditions of failure and restoration of its services (say $S_{7k}...S_{7k+4}$).

Since during prevention it is possible to detect and eliminate not only one but also several vulnerabilities from the set [1...nv], the parameter $\alpha_j$ of the probability of j (j ∈ [1...nv]) detecting vulnerabilities is considered in the model. Values $\alpha_1, \alpha_2, ... \alpha_j, ... \alpha_{nv}$

have a discrete distribution law. As a basis, the geometric law of the distribution of coefficients $\alpha$j with the default parameters p = 0.7 (probability of detecting one vulnerability) and q = 1–p = 0.3 is adopted in the work.

A system of Kolmogorov–Chapman differential equations was constructed for the digraph in Figure 17, in which three blocks were allocated for each fragment of the model.

The value of the availability function is determined by the expression:

$$A(t) = \sum_{i=0}^{k} P_{7i}(t). \tag{16}$$

6.4.3. Research and Analysis of the Results

Input data (values of input parameters) for building a multi-fragment model of the web service are presented in the Table 7.

**Table 7.** The value of the MFM of web services input parameters.

| No. | Parameter | Value | Unit |
|---|---|---|---|
| 1 | Failure rate caused by DNS software design faults | $3 \times 10^{-5}$ | 1/hours |
| 2 | Failure rate caused by Apache HTTP server software design faults | $1.5 \times 10^{-5}$ | 1/hours |
| 3 | Failure rate caused by Oracle WebLogic application server software design faults | $5 \times 10^{-4}$ | 1/hours |
| 4 | Failure rate caused by software design faults of MySQL DBMS | $3 \times 10^{-4}$ | 1/hours |
| 5 | DNS service recovery rate | 1.49992 | 1/hours |
| 6 | Apache HTTP Web Server recovery rate | 1.71420 | 1/hours |
| 7 | Oracle WebLogic Application Server recovery rate | 0.99995 | 1/hours |
| 8 | MySQL DBMS server recovery rate | 1.09085 | 1/hours |
| 9 | Rate of attacks on the vulnerability of the DNS service | $6.3 \times 10^{-3}$ | 1/hours |
| 10 | Criticality of attacks on the vulnerability of the DNS service | 0.77 | 1/hours |
| 11 | The frequency of DNS service restart after an attack | 5 | 1/hours |
| 12 | The probability of eliminating a vulnerability after it appears during an attack | 0.15 | 1/hours |
| 13 | The preventive information security audit test rate | $4.63 \times 10^{-4}$ … $1.14 \times 10^{-4}$ | 1/hours |
| 14 | The recovery after an information security audit rate | 0.33…1 | 1/hours |
| 15 | The number of undetected vulnerabilities in the initial stage | 10 | |

Figure 18 shows the results of single- and multi-fragment models for basic values of input parameters. The following derived values of the resulting indicators were obtained:

- the time when the multi-fragment model starts to show a gain in availability relative to the Markov model, which considers attacks Tup1 = 10,450.7 h.
- the time when the multi-frame model starts meeting the availability requirement of 0.999 Tup2 = 14,949.4 h.
- the greatest decrease in the level of availability of the multi-fragment model relative to the Markov single-fragment model $\Delta$A = 0.0015.

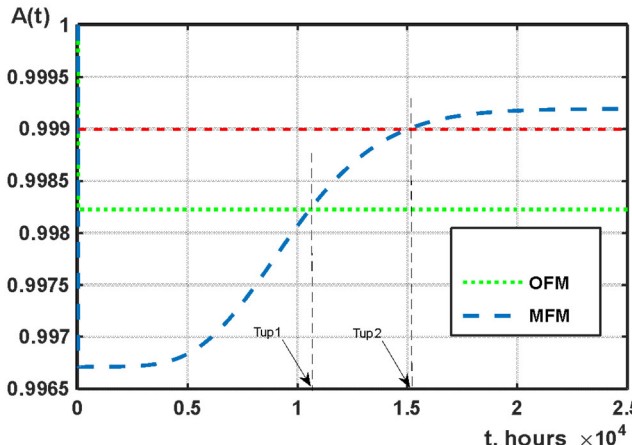

**Figure 18.** The results of the evaluation of the availability of the web service under the condition of the basic values of the parameters.

Analysis of graphs in Figure 19a showed that the value of the parameter $\lambda_{PrT}$ significantly affects the value of the minimum of the availability function. This is explained by the fact that with an increase in the number of preventive security audits, the total time that the system is in a preventive state also increases. The value of the parameter $\lambda_{PrT}$ also affects the speed of crossing the availability function of the given required level of 0.999 (when the intensity of prevention is reduced by 4 times, the time of Tup "exit" to the level of 0.999 increases by 11%). Analysis of graphs in Figure 19b showed that the value of the $\mu_{PrT}$ parameter affects the value of the minimum of the availability function of the multi-fragment model (increasing $\mu_{PrT}$ by 2 times gives an increase in the Amin indicator by 0.07%). Additionally, from Figure 19b, the influence of the $\mu_{PrT}$ parameter on the speed of the intersection with the availability function of the required level of 0.999 can be detected (with a decrease in the time of preventive maintenance, the system will begin to provide the necessary level of availability requirements earlier).

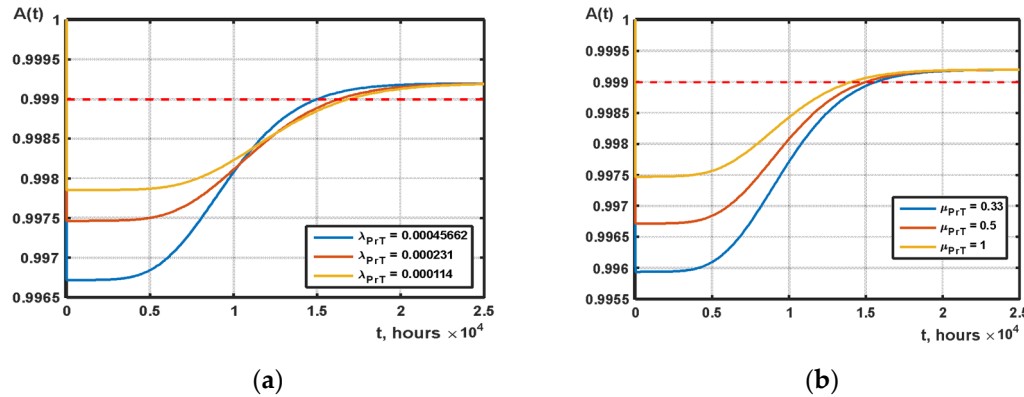

(**a**)                  (**b**)

**Figure 19.** Comparison of the resulting functions of the multi-fragment model at different values of the parameters $\lambda_{PrT}$ (**a**) and $\mu_{PrT}$ (**b**).

Thus, four cases were considered to demonstrate the proposed principles of combined modeling. The investigated CISs illustrate individual cases of application of Markov, multi-fragment, multiphase and semi-Markov models as a single model choice option. The RBD model of the cloud video system is also considered a modeling option based on the principle of component combining. A comparative analysis of the simulation results was performed, which illustrated the increase in the accuracy of the availability function assessment (as an indicator of reliability and cybersecurity) for various options of applying the proposed method.

## 7. Discussion

The suggested strategy for assessing cloud and IoT systems dependability, availability and cybersecurity is based on continuous collecting, comparing (classification), choice and combining Markov and semi-Markov models and can be called a C5 strategy. It provides systematic and step-by-step specifications and builds an adequate and accurate model to evaluate key indicators of CIS. Every "C" is important for implementing the approach because:

- the first one postulates **continuous** evolution of the model(s) together with the evolution of the system caused by (1) changing of its parameters and parameters of the physical and cyber environment, and (2) events that should be considered to assure more accurate calculation or prediction of CIS indicators;
- the second "C" describes the necessity of **collecting** data about faults, failures, vulnerabilities, cyber-attacks, violations of data privacy, applied patches, etc. to obtain as much actual data as possible for the assessment of CIS;
- the third one provides **comparing** and renewed classification of a model set based on analysis of CIS operation (or reengineering) data. It allows actualizing the model base and assuring completeness of the set of models that should be applied for assessment;
- the fourth "C" creates the possibility of **choice** and utilizing "off-the-shelf" models with understandable techniques for their development and implementation;
- the last one provides a composition of different models during the application of CIS by time, component or mixed **combining**.

The C5 strategy is, in some sense, an analogue of the well-known continuous integration/continuous deployment (CI/CD) principle for the methodology of deploying software projects in a cloud environment (DevOps). Principal CI/CD corresponds well to almost all components of C5 and can be refined in some way.

The discussed cases of availability and cybersecurity assessing CISs have demonstrated applying elements of the C5 strategy for different domains.

The first case illustrates the increased accuracy of modeling for an adequate choice of the Weibull distribution law on $2 \times 10^{-5}$. In the second case, simulation results showed that the difference between the cloud system availability indicators determined by RBD and Markov models have differences not exceeding $\Delta A = 0.0034$. Model multi-fragmentation considering the change in the CDN component failure rate caused by attacks allows for increasing the accuracy of the assessment by $3 \times 10^{-3}$.

In the third case, the choice of the mathematical apparatus of a multiphase model is substantiated. The selection method determined the value of the parameter $\Delta T_{VER} = 7$ days (168 h), which allows concluding that the value of the system unavailability function is not more than 0.02 for 2 years (20,000 h) of operation. The fourth case illustrates the increase in the accuracy of the availability indicator assessment by $\Delta A = 0.0015$ relative to the Markov single-fragment model.

More examples can be analyzed in other works [10,48]. These cases allow the conclusion that cybersecurity assessment (integrity and accessibility attributes), first of all, is supplementary to assessing the availability of CIS. Risks of successful cyber-attacks should be analyzed and tolerated in terms of the influence on final availability. Additionally, the proposed principles, algorithms, and models can be used to evaluate CIS privacy, which is the close attribute of cybersecurity, by analysis of component vulnerabilities according to the corresponding privacy attack vector.

## 8. Conclusions and Future Directions

This study was aimed at overcoming the contradictions associated with the use of the mathematical apparatus of Markov and semi-Markov processes for the evaluation of complex systems, which, of course, CISs are. Assessment of dependability and cybersecurity of complex systems, especially CISs, is usually carried out by researches as follows. Researchers who choose Markov models justify their choice over semi-Markov models by

the fact that obtaining reliable values for additional parameters is problematic, therefore, it is better to have a simpler Markov model, but with reliably determined parameters. Researchers, for whom semi-Markov models are more familiar, base their choice on the fact that such models more adequately describe the processes of failures, cyber-attacks, repairs, prevention, etc., without considering the risks of expanding the space of parameters to considered values. Both approaches also do not consider the decrease in the accuracy of the models over time after certain events in the system.

We have substantiated and illustrated the implementation of the approach based on the C5 strategy, which partially overcomes these inconsistencies and introduces a flexible strategy for the selection and combination of models. The considered cases describe the elements of the procedures for the selection and combination of mathematical models of different types of CIS, namely RBD-based, Markov, and semi-Markov models, and their combinations including the application of multi-fragmental and multiphase models. A set of the models can be added to non-Markov models considering specific features of domains, safety issues, etc. [57,58]. It provides an opportunity to make decisions about the building and evolution of the models during the life cycle of a dependable CIS, including the operation stage to assure high accuracy of the assessment.

The novelty of the research results is that the proposed method of CIS assessment is based on the fact that the strategy of model design consists of the step-by-step selection, adaptation, and possible change of the type and parameters during the use of the system. The CIS Markov and semi-Markov models classifier consisting of 52 elements was developed. Using this method, the single and combined models of CIS availability and cybersecurity were investigated. They are based on a theoretical set description of options for combining MMs and SMMs, considering features of key processes, properties and parameters of system and environment. The analyzed relationships between them allow reducing model uncertainty and justifying the choice of means to ensure dependability at various stages of the life cycle. In turn, such an approach increases adequacy and accuracy of assessment, as illustrated in the discussed cases for the different CISs.

It is important to emphasize an additional possibility related to using the proposed approach to increase CIS dependability and cybersecurity assessment trustworthiness. For this, the results of simulations using various methods, including attack trees, simulation modeling, etc., can be used for further comparison and analysis. The limitation of the suggested method is that the combination of models by time and events is described rather heuristically. Analytical solutions are limited by the complexity of the models due to the complexity of the systems. Most likely, it is about the development of a new class of multiphase models with controlled transitions and fitted multiphase. Another limitation, of course, concerns the accuracy of data for parameterization, which influences making decisions about increasing accuracy when experimenting and changing models.

The most important directions for future research are the following:

- development of the framework for C5 approach automation for different types of cloud and IoT systems. The results of the paper (schemes and algorithms) can be used for designing such a framework as a service or embedded technology for online support of operating systems;
- more detailed development of the techniques for combining models and analytical description of the combined once considering different reasons for switching and composition of the models. It requires the specification of all important events and conditions that should be taken into account and strong procedures of implementing options for combining;
- collection of the data for estimation and prediction of CIS models' parameters. It concerns, first of all, information about vulnerabilities and cyber-attacks, and application of ML to calculate parameters;
- the C5 approach should be added by proactive techniques for assessment and assurance of CIS dependability, cybersecurity, privacy, and resilience based on Big Data

analytics [59,60] and machine learning [61] methods to analyze data and support decision-making about choice and combining and recombining models;

- application of this approach for combining hidden MMs and SMMs to assess privacy [62] extending the model base.

**Author Contributions:** Conceptualization, V.K.; methodology, V.K., Y.P., and O.I. (Oleg Ivanchenko) ; models and software, Y.P., O.I. (Oleg Ivanchenko ), and O.I. (Oleg Illiashenko); validation, H.F., O.I. (Oleg Illiashenko); formal analysis, V.K., O.I. (Oleg Illiashenko); investigation, all authors; resources and data curation, Y.P., and O.I. (Oleg Ivanchenko); writing—original draft preparation, all authors; writing—review and editing, H.F., O.I. (Oleg Illiashenko); visualization, Y.P., O.I. (Oleg Ivanchenko); supervision, V.K.; project administration, O.I. (Oleg Illiashenko); funding acquisition, Y.P., O.I. (Oleg Ivanchenko), O.I. (Oleg Illiashenko) All authors have read and agreed to the published version of the manuscript.

**Funding:** This work is funded by the Ministry of Education and Science of Ukraine, project DATIoT (Dependability Assurance methods and Technologies for intellectual industrial IoT systems, N° 0122U001065, 2022-2023).

**Institutional Review Board Statement:** Not applicable.

**Informed Consent Statement:** Not applicable.

**Data Availability Statement:** Not applicable.

**Acknowledgments:** The authors appreciate the staff of the Department of Computer Systems, Networks and Cybersecurity of the National Aerospace University "KhAI" and participants of the monthly scientific seminar "Critical Computer Technologies and Systems" at this Department for invaluable inspiration, hard work, and creative analysis during the preparation of this paper.

**Conflicts of Interest:** The authors declare no conflict of interest.

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
