# Peer review of "Combining Markov and Semi-Markov Modelling for Assessing Availability and Cybersecurity of Cloud and IoT Systems"

_cryptography, doi:10.3390/cryptography6030044_

Round 1

Reviewer 1 Report

The authors present a combining Markov and semi-Markov Modelling for Assessing Availability and Cybersecurity of Cloud and IoT Systems. In this sense, the authors suggest a strategy (C5) for assessing cloud and IoT (CISs) systems’ dependability, availability, and cybersecurity based on continuous collecting, comparing, choice, and combining Markov and semi-Markov models (MM&SMMs).

 In the introduction, I suggest removing the motivation sub-section and including it as part of the introduction. Additionally, in the introduction is necessary to include a paragraph that indicates reliably the main contribution of the paper.

 Section 1.2. State-of-the-Art must not be a sub-section. It must be Section 2 in the paper.

 The state of the art appears to be a list of abstracts. It is necessary to complement it with a comparison among the initiatives analyzed and the proposal of the authors. It may be a comparative table that allows identifying advantages and drawbacks among them.

 Section 1.3. Objectives and structure can be part of the Introduction section.

 Section 2. Approach and stages is clear and concise, however, it requires a conclusion paragraph linking it to the following section.

Section 5.4.3. Research and analysis of the results must be an independent section with information more complete. Section 7 must be Conclusion and Future directions.

The paper presents inappropriate self-citations by authors.

The authors with more self-citations are the follows:

The author Vyacheslav Kharchenko: 14 self-citations. Yuriy Ponochovnyi: 9 self-citations. The author named Oleg Ivanchenko appears 4 self-citations. These practices are not ethical, it is necessary to reduce the number of self-citations at least 2 o 3 self-citations that the authors consider more relevant.

Author Response

Dear Reviewer,

Best regards,

Authors

Reviewer 2 Report

The authors of this paper suggest a strategy for assessing cloud and IoT systems’ dependability, availability, and cybersecurity based on continuous collecting, comparing, choice, and combining Markov and semi-Markov models.

This is a quite interesting paper whose subject is very well presented. Some minor grammatical/syntax errors can be corrected during the preparation of the camera ready version. The reviewer believes that the paper deserves publication.

Author Response

Dear Reviewer,

Best regards,

Authors

Reviewer 3 Report

Cloud and IoT systems are very complex, multi-component, and distributed systems. A certain level of generalization is necessary for the analysis and assessment of reliability and cyber security, which determines the risks to the accuracy of calculating indicators. However, the novelty of the paper is not highlighted. It is difficult to see if the results are new enough.

The paragraph starting from 'The study [18] addressed Markov reliability model for analyzing' should be rewritten. More explanation regarding the methodology is needed.

An important component of the approach also offers the possibility to combine models both in time (time combination, t.c.) and by individual components or subsystems (component combination, c.c.). Therefore, it is about ensuring an adequate selection and "rejuvenation" of the model base of MMs, SMMs, and the use of combined models for the evaluation of cloud and IoT systems. Combining and fitting models in space “time-components” provide increasing adequacy and accuracy of CIS dependability and cyber security assessment. I find this argument is not particularly convincing. Some examples should be discussed. 

The general description of the CSS should be detailed. What is the time complexity of the process proposed?

In equation (5), the exponent is not correct. The parameter should be a random variable itself. Please double-check and correct.

The conclusion is written in a hasty manner. Some future directions and open problems should be discussed. The limitations of the method should be discussed.

Author Response

Dear Reviewer,

Best regards,

Authors

Round 2

Reviewer 1 Report

The authors have addressed all my comments. The paper's content has been significantly improved. The paper has a better structure. I think this paper can be published in its present form.

Reviewer 3 Report

The paper can be accepted now.